



# Rejuvenating the ocean: mean ocean radiocarbon, CO₂ release, and radiocarbon budget closure across the last deglaciation

Luke Skinner[1], Francois Primeau[2], Aurich Jeltsch-Thömmes[3,4],
Fortunat Joos[3,4], Peter Köhler[5], Edouard Bard[6]

[1]Godwin Laboratory for Palaeoclimate Research, Earth Sciences Department, University of Cambridge, Downing Street, CB2 3EQ Cambridge, UK
[2]Department of Earth System Science, University of California, Irvine, California, USA
[3]Climate and Environmental Physics, Physics Institute, University of Bern, Bern, Switzerland
[4]Oeschger Centre for Climate Change Research, University of Bern, Bern, Switzerland
[5] Alfred-Wegener-Institut Helmholtz-Zentrum für Polar-und Meeresforschung (AWI), P.O. Box 12 01 61, D-27515 Bremerhaven, Germany.
[6] CEREGE, Aix Marseille Univ., CNRS, IRD, INRAE, College de France, Technopole de l'Arbois, BP 80, 13545, Aix-en-Provence, France.

*Correspondence to*: Luke Skinner (lcs32@cam.ac.uk)

**Abstract.** Radiocarbon is a tracer that provides unique insights into the ocean's ability to sequester CO₂ from the atmosphere. While spatial patterns of radiocarbon in the ocean interior can indicate the vectors and timescales for carbon transport through the ocean, estimates of the global average ocean-atmosphere radiocarbon age offset (B-Atm) place constraints on the closure of the global carbon cycle. Here, we apply a Bayesian interpolation method to compiled B-Atm data to generate global interpolated fields and mean ocean B-Atm estimates for a suite of time-slices across the last deglaciation. The compiled data and interpolations confirm a stepwise and spatially heterogeneous 'rejuvenation' of the ocean, suggesting that carbon was released to the atmosphere through two swings of a 'ventilation seesaw' operating between the North Atlantic and the Southern Ocean/North Pacific. Sensitivity tests using the Bern3D model of intermediate complexity demonstrate that a portion of the reconstructed deglacial B-Atm changes may reflect 'phase-attenuation' biases that are unrelated to ocean ventilation, and that could arise from independent atmospheric radiocarbon dynamics instead. However, when correcting for such biases, the sensitivity tests further demonstrate that evolving ocean-atmosphere exchange could still account for at least one third of deglacial atmospheric CO₂ rise. Approximately half of the contribution to CO₂ rise appears to have been associated with the Bølling-Allerød, while the rest was linked mainly to Heinrich Stadial 1 and the Younger Dryas. Our global average B-Atm estimates place further new constraints on the long-standing mystery of global radiocarbon budget closure across the last deglaciation and suggest that glacial radiocarbon production levels are likely underestimated on average by existing reconstructions.



**Summary**

Radiocarbon is best known as a dating tool, but it also allows us to track CO2 exchange between the ocean and atmosphere. Using decades of data and novel mapping methods, we have charted the ocean's average radiocarbon 'age' since the last Ice Age. Combined with climate model simulations, these data quantify the ocean's role in atmospheric CO2 rise since the last ice
Age, while also revealing that Earth likely received far more cosmic radiation during the last Ice Age than hitherto believed.

## 1 Introduction

The evolving spatial distribution of marine radiocarbon provides unique constraints on air-sea gas exchange at the sea surface, and the ocean dynamics that convey surface ocean-atmosphere $pCO_2$ differences into the ocean interior (Koeve et al., 2015). These transports of carbon in turn set constraints on the ability of the ocean to store $CO_2$, away from the atmosphere, in the
form of either 'disequilibrium' or 'respired' dissolved inorganic carbon (Galbraith and Skinner, 2020). A slower overturning circulation enhances the accumulation of respired carbon in the ocean interior, in parallel with greater radiocarbon decay (Menviel et al., 2017; Eggleston and Galbraith, 2018; Tschumi et al., 2011). Reduced air-sea exchange impedes the release of respired carbon from upwelled water to the atmosphere, while also impeding radiocarbon input from the atmosphere to the ocean (e.g. Khatiwala et al., 2019; Stocker and Wright, 1996). Both processes enhance carbon sequestration in the ocean
(Galbraith and Skinner, 2020; Eggleston and Galbraith, 2018), while depleting the ocean's average radiocarbon activity relative to the atmosphere (Skinner and Bard, 2022). Therefore, estimates of the global average ocean-atmosphere radiocarbon age offset (B-Atm), may provide powerful constraints on the evolving carbon exchange between the ocean and atmosphere (Siegenthaler et al., 1980). This in turn has implications for the closure of the radiocarbon budget of the atmosphere, given that the atmospheric radiocarbon inventory is largely set by the radiocarbon production rate, and ocean-atmosphere radiocarbon
exchange (Siegenthaler, 1989).

Few estimates of global average B-Atm prior to the instrumental record currently exist. This largely relates to the difficulty of estimating the 'volumetric representativity' of sparse data points, in order to weight their contribution to the global mean. An initial estimate, based on a simple unweighted arithmetic mean of existing data from the Last Glacial Maximum (LGM, ~20 ka) (Sarnthein et al., 2013), demonstrated a significant increase in the mean ocean-atmosphere radiocarbon age offsets (B-
Atm). A subsequent study instead made use of a Bayesian interpolation method, over a 4° resolution global grid, to generate a global field of B-Atm offsets, from which to derive an appropriately volume-weighted global average (Skinner et al., 2017). This alternative approach again showed that the glacial ocean was significantly more radiocarbon-depleted relative to the contemporary atmosphere, as compared to the pre-industrial ocean, especially in the deep Atlantic and Southern Ocean (Sikes et al., 2016a; Skinner et al., 2021; Skinner et al., 2010; Gottschalk et al., 2020). Using yet another approach, whereby individual
data points were weighted according to their location (and their current offset from modern regional/basin averages), a recent study has again confirmed an 'aging' of the deep ocean at the LGM (Rafter et al., 2022). While the latter study chose to focus





on the deep ocean > ~1000m water depth, and therefore did not yield global mean estimates, it is clear that the inferred anomalies for specific depth intervals (corresponding to density classes in the modern ocean) would also require a significant
increase in the global mean B-Atm offset.

Overall, a consistent picture has therefore emerged of a relatively 'aged' dissolved inorganic carbon (DIC) pool in the LGM ocean. A recent review of the existing data has further demonstrated coherent patterns in marine radiocarbon evolution *since* the LGM, across the last deglaciation (Skinner and Bard, 2022). Several key observations have so far emerged from the
compiled data: 1) at the onset of deglaciation, during Heinrich Stadial 1 (HS1, ~17.5-14.7ka), and to a lesser extent the Younger Dryas (YD, ~12.9ka-11.7ka), B-Atm offsets increased in the deep and intermediate North Atlantic, while they decreased in the Southern Ocean and North Pacific (particularly in the intermediate North Pacific, suggesting a 'ventilation seesaw' (e.g. Broecker, 1998; Skinner et al., 2013; Skinner et al., 2014; Menviel et al., 2018; Ahagon et al., 2003; Menviel et al., 2014; Freeman et al., 2015; Max et al., 2014; Okazaki et al., 2010; Walczak et al., 2020); 2) B-Atm offsets decreased significantly
throughout the global ocean at the Bølling-Allerød (BA, ~14.7-12.9ka), in places suggesting an overshoot to values younger than pre-industrial (e.g. Barker et al., 2010); and 3) surface reservoir ages in the high latitudes of the North Atlantic and Southern Ocean tend to track B-Atm changes at intermediate depths (Skinner et al., 2019), suggesting limited changes in transport <2km water depth (and therefore a significant contribution from restricted gas-exchange efficiency), but with evidence for additional transport pathway and/or rate changes in the deepest ocean >2km (Skinner et al., 2021; Marchal and
Zhao, 2021a). Encouragingly, similar observations have recently been reported based on a complex approach to data screening, averaging, and weighting (Rafter et al., 2022). The latter study emphasized the influence of transport changes on B-Atm offsets, inferring enhanced deep-water formation in the North Pacific during the LGM (reaching mid-depths), despite a reduction of transport rates across the Pacific at the LGM, and in the deep North Atlantic during HS1 and the YD.

Several questions arise in light of the existing body of deglacial marine radiocarbon data. First, what do the existing data imply for *global average* (i.e. rather than regional, or deep-ocean) B-Atm offsets across the last deglaciation? Secondly, can we determine the degree to which observed changes in the mean ocean B-Atm offset reflect changes in 'ventilation' specifically (i.e. circulation rates *and* gas exchange), as opposed to independent changes in atmospheric radiocarbon for example (Franke et al., 2008)? Thirdly, can we quantify the likely impact of past radiocarbon ventilation effects (regardless of their origin) on
ocean-atmosphere carbon partitioning, and therefore atmospheric $CO_2$ (Skinner and Bard, 2022)? Finally, can the temporal evolution of mean ocean B-Atm offsets be reconciled with past atmospheric radiocarbon activities, and past radiocarbon production rates; and do they resolve the long-standing 'mystery' of deglacial radiocarbon budget closure (Broecker and Barker, 2007; Köhler et al., 2022; Kohler et al., 2006)?

Here, we attempt to address each of these questions in turn. We update a previously deployed Bayesian interpolation technique (Skinner et al., 2017), which we apply to an updated compilation of radiocarbon data spanning the last deglaciation, to derive





estimates of global average B-Atm offsets. These estimates are interpreted against a suite of new sensitivity tests and transient simulations, conducted using the Bern3D model of intermediate complexity (Müller et al., 2006; Ritz et al., 2011; Roth et al., 2014). The new simulations are aimed at constraining the potential magnitude of 'ventilation' *versus* 'non-ventilation' effects

in global mean B-Atm offsets, as well as associated changes in ocean-atmosphere $CO_2$ partitioning.

## 2 Methods

### 2.1 Radiocarbon data

Our data compilation is based on the work of Zhao et al. (2018), as presented in Skinner and Bard (2022). The compilation has been reconciled with a similar collection of data that has recently emerged (Rafter et al., 2022). A key difference in our

compilation is that we include warm water (near surface) coral data, and surface reservoir age data where direct estimates exist (e.g. Skinner et al., 2019). Our compilation adopts revised chronologies and radiocarbon age offsets consistent with the latest *Intcal20* (Reimer et al., 2020) and *Marine20* (Heaton et al., 2020) calibration curves. To update the chronologies of the collected time-series, where the only age control derives from planktonic radiocarbon dates, we have used the *Marine20* calibration curve (Heaton et al., 2020), in conjunction with modern local deviations from the global average surface reservoir

age (ΔR values ~ local R-age – 550) (Heaton et al., 2020), to generate 'reservoir-age' corrected calibrated calendar ages, using the R package *Bchron* (Parnell et al., 2008). While this approach has the advantage of remaining as faithful as possible to the original published age-scale for each record, it also has the notable drawback of neglecting potential changes in ΔR, which are known to have occurred in mid/high latitudes (e.g. Skinner et al., 2019), and that have been directly estimated in some of the studies included in the compilation. For records updated in this way, where the density and/or down-core variability of

planktonic radiocarbon dates in a sediment core is sufficient to result in age-reversals (i.e. older dates stratigraphically above younger dates), we have generated new sediment depth-age models using the MCMC approach of *Bchron*, from which maximum likelihood calendar age estimates and 95% uncertainty intervals (~2 sigma) are obtained for each sample depth. For time-series that have made use of U-series dating (e.g. Robinson et al., 2005; Burke and Robinson, 2012; Chen et al., 2015; Hines et al., 2015), or stratigraphic alignments (including stratigraphic alignments that have provided time-varying surface

reservoir age corrections, (e.g. Skinner et al., 2010; Gottschalk et al., 2020; Austin et al., 2011; Peck et al., 2006); as well as *a priori* assignment of down-core changes in reservoir ages (e.g. Ronge et al., 2020), we adopt the published calendar ages, as these do not depend on the atmospheric radiocarbon calibration curve.

For all compiled data, we have recalculated radiocarbon reservoir age offsets relative to the *Intcal20* (Reimer et al., 2020)

atmospheric reference curve using the *R* package *Radcal* (Soulet, 2015). This approach yields 95% uncertainty limits for the resulting radiocarbon age offsets based on the joint calendar age and radiocarbon age uncertainties. The resulting probabilistic B-Atm estimates differ from simple differences between marine and atmospheric radiocarbon values to some extent (Rafter et al., 2022). Estimates of the relative (radiocarbon) isotopic enrichment of the ocean *versus* the atmosphere are expressed as radiocarbon age offsets between the ocean and the contemporary atmosphere (i.e. in [14]C years), which are equivalent to



$-8033 \times \ln (\alpha_{O-Atm})$, where $\alpha_{O-Atm}$ represents the ratio of marine radiocarbon activity *versus* the contemporary atmospheric radiocarbon activity at time T, i.e. $Fm_O^T/Fm_{Atm}^T$ (Soulet et al., 2016). We do not refer to offsets between marine and atmospheric $\Delta^{14}C$ (i.e. $\Delta^{14}C_O$-$\Delta^{14}C_{Atm}$), since this metric does not relate to the isotopic depletion of the marine reservoir relative to the atmosphere in a predictable way without knowledge of the absolute atmospheric and marine radiocarbon activities (e.g. Cook and Keigwin, 2015).


Data quality flags are assigned to anomalous values or datasets, including those that yield negative ventilation ages, negative reservoir ages, or negative benthic-planktonic offsets (B-P), or that exhibit significant down-core age-reversals (e.g. Rose et al., 2010). In addition, datasets that deviate significantly from regional trends, for example as a result of alternative age-models, are also flagged. The latter category includes records based on 'plateau tuned' (PT) chronologies (Sarnthein et al., 2015; 145 Sarnthein et al., 2007; Sarnthein et al., 2020; Sarnthein et al., 2013; Ausín et al., 2021) that differ significantly from alternative reconstructions using the same data, or from reconstructions at proximal sites (Skinner and Bard, 2022). Such differences may relate to identifiable drawbacks of the 'plateau tuning' methodology (Bard and Heaton, 2021). Datasets that are interpreted as being influenced by hiatuses are also flagged (e.g. Ronge et al., 2020; Ronge et al., 2019), as are data that are interpreted as having been influenced by localized geological or sedimentary carbon sources (Rafter et al., 2019; Rafter et al., 2018; Lindsay 150 et al., 2016; Lindsay et al., 2015; Ronge et al., 2016; Bova et al., 2018; Stott et al., 2009; Marchitto et al., 2007). A similar selection was effectively made in the recent study of (Rafter et al., 2022), where all data from water depths less than ~1000m were omitted from consideration, and where individual data points were removed from regional water-depth groupings based on their deviation (by > 3 sigma) from the group's weighted arithmetic mean.

Although the data compiled in this study have all been published, including the vast majority in a previous compilation (Zhao et al., 2018), it remains possible that some data have been affected by as yet undetected biases arising from bioturbation (Bard et al., 1987; Dolman et al., 2021), and/or diagenesis (Wycech et al., 2016). Unless obviously biased signals are produced (Missiaen et al., 2020), such biases can be extremely difficult to positively identify, but could have profound impacts (Lougheed et al., 2020). This presents a significant challenge. However, in order to limit the potential impacts of anomalies 160 caused by bioturbation (Dolman et al., 2021), sediment cores with average accumulation rates < 2cm/kyr have been flagged and set aside from our interpolations. Of the retained data, 75% derive from sites with accumulation rates > 10 cm/kyr and 95% derive from sites with accumulation rates > 4cm/kyr. A comparison of late Holocene data (< 6 ka BP) with pre-industrial observations (Key et al., 2004) serves to demonstrate the general fidelity of B-Atm offsets derived from fossil substrates, as compared to modern seawater values (see **Figure 1**, $R^2$ = 0.81). However, this comparison also illustrates the significant scatter 165 that exists in the fossil data (RMSE ~ 273 years; or an 'inverse prediction interval', more appropriate if considering a 'calibration' of fossil data, ~ RMSE x 1.96 = 535 yrs (McClelland et al., 2021)). A similar result has recently been demonstrated for sub-modern B-Atm offsets by Rafter et al. (2022). The observed scatter is expected to derive primarily from sedimentary and sampling issues, rather than analytical uncertainty, and calls for caution when interpreting the subtler features that emerge



from the collected data. Accordingly, an underlying premise of our work (as for previous compilations) is that the temporal
trends and spatial patterns that emerge from a large body of independent data are far less likely to be dominated by sedimentary
or diagenetic biases that can more readily influence individual sediment cores (Ronge et al., 2019; Stott, 2020).

In order to assess the implications of applying the data flags described above, three alternative interpolations were performed:
1) the baseline interpolation, where all flagged data were excluded (**Table 1**, 'baseline'); 2) as for the baseline, but with low-
sedimentation rate sites (<2cm/kyr) retained (**Table 1**, 'low sedimentation'); and 3) as for the baseline, but also omitting a
single study from the sparsely sampled deep Indian Ocean, where sedimentation rates were slightly above the 2 cm/kyr cut-
off (**Table 1**, 'Indian variant'). Ultimately, these alternative approaches to data flagging have only a minor impact on the
resulting global averages, though the inclusion/exclusion of specific data points from sparsely sampled regions (e.g. for the
'Indian variant' scenario) can significantly influence the inferred spatial patterns (see below, and **Figure 6**). Notably, in
keeping with a similar assessment made by Rafter et al. (2022), we also find that the inclusion of low sedimentation sites has
no major effect on our findings, particular for the global averages (**Table 1**).

## 2.2 Time-slices

We isolate data associated with six key time-slices, based on their calendar ages and associated uncertainties: the LGM (19-
21.8 ka BP), Heinrich Stadial 1 (15-17.5 ka BP), the Bølling-Allerød (BA, 12.8-14.8 ka BP), the Younger Dryas (YD, 11.8-
12.7 ka BP), the early Holocene (EHOL, 9-11 ka BP), and the late Holocene (HOL, <6 ka BP). Note that the boundaries of
these time intervals are informed by, but do not correspond precisely to their respective ice-core chronozone definitions
(Rasmussen et al., 2014), due to the generally lower resolution of the marine records, and the need to be pragmatic in avoiding
as far as possible misattributing data to an adjacent chronozone. Where calendar age uncertainties result in an ambiguous
assignation of a single time-slice, no time-slice is assigned.

## 2.3 Radiocarbon 'ventilation' metrics

We note that the term 'ventilation' is defined here as comprising the processes that influence physical and chemical property
exchanges between the ocean interior and the atmosphere (Skinner and Bard, 2022). Accordingly, the term '(radiocarbon)
ventilation age' is not used to refer to 'ideal ages' or 'transit times' (England, 1995; Marchal and Zhao, 2021b), but rather to
the degree of isotopic ($^{14}$C/C) disequilibrium between the ocean and atmosphere, as determined by gas exchange efficiency
and water transport rates (Koeve et al., 2015). In this context 'gas exchange efficiency' is a catch-all term that refers primarily
to the gas transfer piston velocity and includes the influence of the mixed layer equilibration time, which in turn depends on
the mixed layer depth for example.

'Radiocarbon ventilation' must in turn be differentiated from measures of *relative isotopic enrichment*, such as B-Atm age
offsets (Soulet et al., 2016). While 'ventilation' refers to a set of processes (principally gas exchange efficiency and water





transport), observation-based metrics such as B-Atm reflect measures of radiocarbon isotopic disequilibrium, typically between the ocean interior and the atmosphere or mixed layer (Skinner and Bard, 2022). This terminology mirrors a standard

geological principle of separating descriptive names (e.g. diamicton) from the inference of process (e.g. glacial till).

In summary, B-Atm should not be used to directly imply transit times or ideal ages, which are not equivalent to 'radiocarbon ventilation ages'. Furthermore, metrics such as B-Atm offsets are influenced by a wider variety of processes than just ventilation (i.e. more than just gas exchange and transport), including e.g. atmospheric radiocarbon production changes

(Heaton et al., 2021). Therefore, B-Atm offsets should only be used to refer to ventilation (i.e. gas-exchange and transport) with caution, as we explore in this study.

## 2.4 Interpolation

We use the interpolation method described in (Skinner et al., 2017), updated to weight input data according to their reported

uncertainties. The method interpolates the observed ocean-atmosphere radiocarbon age offset anomalies onto the grid of an ocean model with 24 vertical levels and $2° \times 2°$ horizontal resolution. The anomalies are defined relative to the modern GLODAP radiocarbon ages (Key et al., 2004). The interpolating function is constructed using a weighted superposition of radial basis functions centred at the sediment-core locations. These basis functions are built here using an exponential basic function, $\varphi(r) = e^{-\varepsilon r}$, with shape parameter, $\varepsilon$. The distance $r$ is defined as the time for tracer impulses to spread out from

the core locations to the rest of the grid, based on the modern transport and density stratification. The basis-function weights, the shape parameter, and two hyper-parameters that scale the relative size of the precisions in the prior and likelihood functions, are inferred using a three-level Bayesian procedure (further details are given in Skinner et al. (2017)). The interpolation process is repeated independently for each time-slice so that the data from one time-slice do not influence the interpolation for the other time-slices.


This interpolation method is somewhat conservative because it is based on the distribution of relative tracer transport/diffusion timescales in the modern ocean. This choice is premised on the need to obtain an interpolated solution in three dimensions that is physically sensible and that is informed by large-scale oceanographic requirements, including vertical stratification, basin margins/topographic boundaries, and dominant transport pathways (e.g. the Antarctic Circumpolar Current, Deep Western

Boundary currents, etc.). It is important to note that the interpolation method does not impose the modern transport on the resulting interpolated fields, nor does it assume strict adherence to the modern density field. Rather, it represents a physically guided and data-constrained anomaly relative to the modern state. The inferred anomalies relax to zero in the absence of data constraints, though some locations in the ocean can have far reaching influence. This method represents a counterpoint to the approach recently taken by Rafter et al. (2022) for example, where spatial interpolations were performed on data projected

onto two-dimensional zonal sections. In the latter study, averages were also produced for time-series grouped and weighted according to their position within the modern density- and B-Atm distributions (i.e. locations closest to the modern mean for



a given basin and density class, are weighted to dominate the mean in the past). With this approach, the spatial distribution of density classes is assumed invariant over time, despite the inference of circulation changes. This mirrors to some extent our use of a modern transport field as a guide to the volumetric representativity of individual sample locations. Interpolating a

sparsely sampled 3D field is a difficult problem, and a diversity of approaches is surely useful; however, it should be noted that (Rafter et al., 2022) did not produce or discuss global averages did not consider the upper ~1,000m of the ocean.

In order to assess the accuracy of the interpolation method of (Skinner et al., 2017), and its ability to represent different circulation states than the modern, we apply it to modelled radiocarbon fields produced in this study using the Bern3D model

(see below), and by (Menviel et al., 2017) and (Menviel et al., 2018) using the LOVECLIM model. We extract data from the modelled fields at the same locations as the available proxy data (e.g. for the LGM), and seek to reproduce the modelled global field using the interpolation method. As shown in **Figure 2**, the interpolation method is generally successful in reproducing the simulated radiocarbon fields for altered circulation states. The interpolation does best for circulation states characterised by altered wind, diffusivity, and gas exchange, yielding average errors for 1000 randomly selected individual grid cell estimates

of RMSE ~ 276 years (INT_ALL60, this study), RMSE ~ 244 years (INT_ALL80, this study), RMSE ~ 398 years (V3LNAw, (Menviel et al., 2017)), and RMSE ~ 511 years (V3LNAwSOwSHWw, (Menviel et al., 2017)) (see **Figure 2** and **Figure 3**). The interpolation performs slightly less well for extremely different circulation states, e.g. yielding RMSE ~ 457 years for the HS1 simulation of (Menviel et al., 2018), and RMSE ~577 years for the collapsed AMOC simulation of (Menviel et al., 2017), both of which yielded reversed radiocarbon gradients in the Atlantic as compared to modern (**Figure 3**). Notably, the

interpolation method reproduces the modelled fields more accurately than model simulations of the LGM are typically able to match observations (Muglia et al., 2018; Menviel et al., 2017).

In the present study, the interpolation is primarily intended for calculating global average B-Atm offsets, using a relatively sparse dataset (N = 124 to 476). It is therefore of particular importance that the interpolation is found to reproduce modelled

global averages even more accurately than the spatial distributions, with RMSE ~ 53 years (**Figure 4**, circles). In contrast, simple arithmetic means of B-Atm values drawn from a sub-set of locations in the modelled fields consistently underestimate the true global mean, by ~274 years on average (**Figure 4**, triangles). This comparison underlines the importance of applying a spatially resolved volumetric weighting to the observations for the derivation of accurate global averages. While our interpolation approach leaves room for improvement, it complements alternative data-constrained modelling approaches that

have been applied to the LGM and the YD for example (Pöppelmeier et al., 2023), and it represents a first step towards addressing the problem of deriving 3D global fields, and appropriately weighted global averages, from sparsely sampled proxy data, without the use of forward or inverse models.

## 2.5 Modelling



The intermediate complexity Bern3D v2.0 Earth System model (Ritz et al., 2011; Roth et al., 2014) has been used to explore a series of idealised scenarios that aim to probe the parallel sensitivities of atmospheric $CO_2$ and marine radiocarbon, subject to changes in global vertical diffusivity, Southern Ocean winds, and/or Southern Ocean gas-exchange efficiency. The implementation of radiocarbon in the Bern3D model is described elsewhere (Müller et al., 2006; Muller et al., 2008; Dinauer et al., 2020), and the simulations presented here extend those performed by (Jeltsch-Thömmes et al., 2019) and (Dinauer et al.,

2020), as well as those performed using an earlier version of the Bern3D model by (Tschumi et al., 2011). The applied version of the Bern3D model comprises a single-layer energy-moisture balance atmosphere with a thermodynamic sea-ice component (Ritz et al., 2011), coupled to a 3D geostrophic-frictional balance ocean (Edwards et al., 1998; Müller et al., 2006), and a 4-box representation of the land-biosphere that simulates the dilution of an atmospheric isotopic perturbation by the land biosphere, but here does not address changes in land carbon stocks (Siegenthaler and Oeschger, 1987). Furthermore, marine

sediments were not implemented in the model set-up used here, as we seek to isolate the impacts of ventilation processes alone (i.e. gas exchange and transport).

A first set of sensitivity tests (PI-), performed under 'pre-industrial' conditions, and run out for 4000 years with a fully interactive climate-carbon cycle system (minus sediments), consisted of step changes in: 1) global vertical diffusivity in the

ocean (Kv); 2) Southern Ocean wind stress (SW); 3) Southern Ocean gas-exchange efficiency, or piston velocity (SG); or 4) a combination of Kv, SW and SG (ALL). In each case the relevant parameter was reduced by 20%, 40%, 60% or 80% relative to its baseline value.

In an additional set of simulations, we employed a range of glacial/deglacial scenarios, similar to (Jeltsch-Thömmes et al.,

2019), where each is again defined by idealised adjustments to vertical diffusivity, wind stress, Southern Ocean gas exchange or a combination of these, as described for the above PI- sensitivity tests. For these simulations, the model was 'spun up' into equilibrium over 35,000 model years under pre-industrial (1700 CE) boundary conditions. The model was then ramped linearly over 5,000 years from the resulting PI equilibrium state into 'glacial' boundary conditions (i.e. representative of 50 ka BP) for ice sheet albedo, greenhouse gas radiative forcing, and insolation, as well as prescribed values for diffusivity, wind stress, and

Southern Ocean gas-exchange efficiency as described above. After this, ice-sheet albedo greenhouse gas radiative forcing, and insolation were varied based on observations since 50 ka BP, with diffusivity, wind stress, and gas-exchange efficiency remaining constant for 32,000 years and then relaxing linearly back to PI control values from 18 ka BP to 14.8 ka BP, after which PI control values were maintained.

In one set of 'glacial/deglacial' model simulations (INT-), atmospheric radiocarbon was prescribed from 50 ka BP based on the *Intcal20* reference curve (Reimer et al., 2020). Radiocarbon concentrations in the model were therefore required to be consistent with the observed atmospheric radiocarbon concentration, via changes in the global radiocarbon inventory (these are equivalent to *ad hoc* changes in radiocarbon production). In a second set of simulations (FIX-), atmospheric radiocarbon





activity was held constant at 140 permil, while in a third set of simulations (CONST-) atmospheric radiocarbon production
rates were held constant at the PI value. In each case a control simulation was performed with evolving boundary conditions
as described above. A series of additional simulations were then performed with altered global vertical diffusivity (Kv),
Southern Ocean wind stress (SW), Southern Ocean gas-exchange efficiency (SG), or a combination of these (see **Table 2** and
**Table 3**). The goal of these transient simulations is not to reproduce the last deglaciation, but to assess the sensitivity of both
marine radiocarbon and atmospheric $CO_2$ to a variety of changes in ocean ventilation (in terms of both the type of forcing, and
its magnitude), including situations where concurrent radiocarbon production rate changes are either minimised (CONST-
/FIX-) or maximised (INT-).

### 3 Results

**Figure 5** illustrates zonally averaged radiocarbon age offsets (B-Atm) for the LGM, HS1, BA, and YD, in the Atlantic and
Pacific basins, based on global interpolations of the compiled data (using the 'baseline' data flags). Differences between each
successive time-slice interpolation are shown in **Figure 6**. Zonal averages and offsets for the Indian Ocean in **Figure 7**. Zonal
averages for the EHOL and HOL, which differ only slightly from the modern field (Key et al., 2004), are included in an
appendix for completeness. As noted elsewhere (Skinner and Bard, 2022; Rafter et al., 2022), the current paucity of data from
the Indian Ocean makes interpolations for this basin somewhat tentative, and more strongly dependent on individual data
points. **Figure 7** illustrates the impact of including a few data points from the LGM and HS1 (Bharti et al., 2022), where
sedimentation rates were just above the 2cm/kyr cut-off threshold. While the intensity of the LGM and HS1 anomalies in the
Indian Ocean are significantly affected by these data, the associated global average B-Atm offsets remain similar (**Table 1**),
albeit with the most significant impact on the reconstructed anomaly between HS1 and the LGM. This comparison highlights
the Indian basin as an important target for future work. It also emphasizes that the global anomaly reconstructed for HS1 is
likely the most uncertain, with a spread of 145 [14]C years between the three scenarios (compared to 50, 65, 38, 4, and 15 [14]C
years for the LGM, BA, YD, EHOL and HOL respectively). Across all the time-slice interpolations, correlations between
observed and interpolated values over the global domain yield $R^2$ values range from 0.59-0.86. Because the interpolation is
not performed on a 2D zonal plane, local B-Atm estimates may deviate from the zonal average where zonal gradients exist.
Furthermore, the interpolations necessarily provide an average 'snapshot' for an entire time-slice, and therefore will mask
variability within the time-slice. These aspects appear to be particularly relevant for HS1 in the Atlantic, as discussed below.

The global interpolations illustrated in **Figure 5** capture the main features of deglacial B-Atm evolution that have previously
been identified in individual time-series, and in compiled time-series that were grouped by basin and region/water depth
(Skinner and Bard, 2022). Broadly similar patterns have also been identified in 2D zonal projection contour plots of compiled
data (Rafter et al., 2022). The main features include:





1) increased B-Atm offsets throughout the global ocean at the LGM, as compared to the modern state (**Figure 5g, h**), in particular >2000m water depth (as demonstrated previously (Skinner et al., 2017), but with a slightly larger anomaly ~800 $^{14}$C years)

2) a step-wise 'rejuvenation' of the ocean interior across the last deglaciation;

3) evidence for positive B-Atm anomalies in the deep North Atlantic, from the LGM to HS1, and again from the BA to the YD, occurring in parallel with changes of broadly opposite sign in the Southern Ocean and the intermediate depth North Pacific (boxed areas in **Figure 6g, h** and **Figure 6c, d**);

4) a marked 'rebound' to lower B-Atm offsets throughout the ocean, from HS1 to the BA (**Figure 6e, f**); and

5) a further rebound to lower B-Atm offsets in the deep North Atlantic, from the YD to the EHOL, again with evidence for changes of opposite sign in the Southern Ocean and intermediate depth North Pacific (boxed areas in **Figure 6a, b**).

Perhaps the most striking aspect of the successive time-slice reconstructions shown in **Figure 5** and **Figure 6** is the marked

drop in B-Atm offsets that coincides with the transition from HS1 to the BA, which involved positively correlated changes in B-Atm in the deep Southern Ocean and deep North Atlantic (Skinner et al., 2013), and which has been linked to an 'overshoot' in B-Atm offsets at some locations (Barker et al., 2010; Hines et al., 2015). Notably, the step change at the BA resulted in a global B-Atm distribution very similar to the modern (**Figure 5c, d**), despite representing the approximate temporal mid-point of deglaciation.


The features identified above can also be discerned in regional time-series averages (Skinner and Bard, 2022; Rafter et al., 2022), e.g. using cubic splines for North Atlantic, Southern Ocean and North Pacific data grouped according to the boxed areas highlighted in **Figure 6**. Regional average splines are illustrated in **Figure 8**, along with the global average B-Atm values that are derived for each successive time-slice (**Table 1**, filled circles in **Figure 8d**). During HS1 and the YD, the regional splines

exhibit broadly anti-phased trends in B-Atm between the deep North Atlantic and the deep Southern Ocean and intermediate North Pacific (shaded vertical bars, **Figure 8**). The collected time-series thus support the broadly antiphase patterns apparent in **Figure 6** (boxed areas). However, during the BA, all three of these regions exhibit B-Atm offsets at least as low as modern, resulting in global average B-Atm that also approaches the modern, and slightly 'overshoots' relative to the subsequent YD (see **Figure 8d**). Consistent with the fact that the Southern Ocean ventilates ~58% of the ocean interior (Primeau, 2005),

**Figure 8** also shows that the global average B-Atm generally tracks the Southern Ocean, while quite different patterns of variability are expressed in the North Atlantic, and the intermediate North Pacific, where more rapid and regionally important fluctuations are apparent (Freeman et al., 2015).

**4 Discussion**





In principle, evolving large-scale patterns of ocean-atmosphere radiocarbon age offsets (B-Atm) will primarily reflect the combined influences of 1) radiocarbon production, 2) ocean transports, 3) air-sea gas exchange efficiency (especially in the regions of deep-water export), and 4) the changing contributions of different source regions to locations in the ocean interior. While the influence of atmospheric radiocarbon production changes on evolving B-Atm offsets is often ignored, it may in principle influence deep ocean B-Atm offsets through relatively rapid (i.e. sub-millennial) changes in atmospheric radiocarbon

activity that are only conveyed to the deep ocean on the millennial time-scale of ocean turn over (Adkins and Boyle, 1997). This can produce a convergence or divergence of marine and atmospheric radiocarbon ages, due to atmospheric radiocarbon changing quickly and the deep ocean remaining relatively invariant (Franke et al., 2008; Heaton et al., 2021).

Below, we discuss how all these processes have influenced marine radiocarbon cycling across the last deglaciation.

Accordingly, we address: 1) evidence for the operation of a 'ventilation seesaw' that we emphasize was linked to *both* gas-exchange and transport anomalies emanating from the main regions of deep- and intermediate water formation (i.e. the North Atlantic, Southern Ocean, and North Pacific); and 2) the potential for atmospheric radiocarbon dynamics (independent of ocean ventilation) to bias B-Atm offsets by ~ hundreds of $^{14}$C years, in particular during the apparent BA 'overshoot'. We further seek to quantify the likely carbon cycle impacts associated with the observed global average B-Atm changes, and to

reconcile these with the evolution of the global radiocarbon budget since the last glacial period.

### 4.1 Ventilation 'seesaws': gas-exchange and transport effects

A survey of the modern ocean's transport pathways indicates that ~86% of the ocean interior is sourced by water that last made contact with the atmosphere in three key regions: the Southern Ocean (contributing ~58%), the high latitude North Atlantic

(contributing ~21%), and the North Pacific (contributing ~7%) (Primeau, 2005). These three regions account for ~31% of the ocean's surface, and 'ventilate' ~86% of the ocean's interior (Primeau, 2005). Although the long equilibration time for dissolved $\Delta^{14}$C(DIC) means that a water parcel's point of last contact with the atmosphere will not necessarily be equivalent to its point of 'radiocarbon equilibration' (high latitude sources are typically characterised by distinct levels of disequilibrium (Bard, 1988; Matsumoto, 2007)), changes in the three main regions of deep-water export will exert a strong influence on

temporal variations in the ocean interior's radiocarbon distribution.

This expectation is clearly borne out in the global interpolations shown in **Figure 5** and **6**, and in the regional stacks illustrated in **Figure 8**. Deglacial changes in marine radiocarbon distribution were apparently dominated by anomalies extending from the North Atlantic (affecting the deep Atlantic in particular), coordinated with anomalies of broadly opposite sign originating

in the Southern Ocean and North Pacific (**Figure 6** and **Figure 8**). The time-slice interpolations therefore cohere with numerous previous proposals for 'ventilation seesaws' operating between the North Atlantic and the Southern Ocean (Broecker, 1998; Skinner et al., 2013; Skinner et al., 2014; Menviel et al., 2018), and between the North Atlantic and North Pacific (Menviel et al., 2014; Freeman et al., 2015; Max et al., 2014; Okazaki et al., 2010; Walczak et al., 2020).



As suggested previously (Skinner et al., 2019; Skinner and Bard, 2022), **Figure 8** also shows that B-Atm offsets in the deep (>2km) North Atlantic (**Figure 8b**, blue line and shaded area) exhibit a similar pattern of variability to the upper ocean (<2km) and surface 'reservoir ages' in the region (**Figure 8b** grey lines with shaded area, and dashed red lines). The same is apparent in the Southern Ocean (**Figure 8d**). These relationships, and their further link to polar climate variability (**Figure 8a, e**), suggest a mechanistic link between the observed deglacial B-Atm variability and the 'thermal bipolar seesaw' (Stocker and

Johnsen, 2003; Epica Community Members, 2006). This association most likely operated via coordinated changes in North Atlantic and Southern Ocean convection and advection (Broecker, 1998; Menviel et al., 2015; Skinner et al., 2014; Skinner et al., 2020), associated with regional changes in sea-ice (Skinner et al., 2019; Rae et al., 2018), winds (Sikes et al., 2016b; Menviel et al., 2018), and/or buoyancy forcing (Ferrari et al., 2014; Hines et al., 2019; Watson et al., 2015). The further coupling between the Southern Ocean and North Pacific has been proposed to relate to freshwater balance in the Pacific basin

(Menviel et al., 2014), possibly influenced by changing moisture transports across the Isthmus of Panama (Leduc et al., 2007), and/or Cordilleran ice mass balance (Walczak et al., 2020).

While the 'ventilation seesaws' noted above may reflect changes in ocean circulation to some extent, it is important to note that they also reflect changes in gas-exchange efficiency. This is demonstrated by the fact that the amplitude of B-Atm

variability in the shallow Atlantic < 2 km water depth (e.g. Freeman et al., 2015; Chen et al., 2015) differs very little from that of surface 'reservoir ages' in the Northeast Atlantic (Skinner et al., 2019) (**Figure 8b**, grey line and dashed red line, respectively). This implies a muted contribution from flow speed changes in the upper Atlantic < 2km (Bradtmiller et al., 2014), and a dominant influence on radiocarbon signatures from gas-exchange (i.e. 'pre-formed ages') instead. Again, the same is true for the Southern Ocean, where B-Atm offsets from the shallow ocean (< 2km) generally overlap with surface

reservoir age reconstructions (**Figure 8d**, grey line and dashed red line, respectively). Accordingly, B-Atm values at the LGM in mid-depths may indeed have been consistent with the modern transport (Marchal and Zhao, 2021a).

In contrast however, 'pre-formed ages' cannot account for the amplitude of B-Atm changes observed in the deep Atlantic (**Figure 8b**, blue line), and the deep Southern Ocean (**Figure 8d**, blue line), indicating a more significant contribution from

water sourcing and/or transport changes >2 km, and perhaps between 2 and 3km water depth in particular (Skinner et al., 2021; Lund et al., 2015; Rafter et al., 2022). The patchy response seen in the zonal average anomaly for the North Atlantic between HS1 and the LGM (**Figure 6h**) contrasts with the clearer signal seen in individual and collected time-series (**Figure 8b**). In part, this likely reflects a spatially heterogeneous hydrographic response, both in the depth domain (Skinner et al., 2021; Lund et al., 2015), and in the eastern *versus* western North Atlantic (Gherardi et al., 2005; Ng et al., 2018). This interpretation is

supported by the less ambiguous positive B-Atm anomaly seen in the deep North Atlantic regional time-series spline during HS1 (**Figure 8b**, blue line and shaded area; see also Rafter et al. (2022)).



The patchy spatial interpolation outcome for HS1 may also reflect a more complex pattern of ventilation during HS1 than is captured by a 'collapsed AMOC' scenario that lasted until the onset of the BA, as indicated by a recent multi-proxy and
modelling study (Pöppelmeier et al., 2023). Furthermore, in addition to tentative evidence for a 'mid-HS1 B-Atm minimum' (Skinner and Bard, 2022), there is also evidence for a drop in B-Atm offsets at intermediate depths (< 2.5km) in the western Atlantic late in HS1 (Robinson et al., 2005; Thiagarajan et al., 2014), and evidence for declining B-Atm offsets in the Nordic Seas ~400 years prior to the onset of the BA (Muschitiello et al., 2019). Any such changes during HS1 will have been averaged out in our time-slice interpolation for 15-17.8 ka BP. These observations underline the need for further detailed reconstruction
of the spatial expression and temporal evolution of ocean-atmosphere radiocarbon offsets across the North Atlantic during HS1, particularly with a view to disentangling transport and gas-exchange impacts.

Overall, the message that emerges from the collected data is that deglacial marine B-Atm changes throughout the global ocean were significantly influenced by *both* air-sea gas exchange effects and mass transport effects (Koeve et al., 2015), with the
latter primarily affecting parts of the deep ocean >2km, but with evidence for a complex response during HS1. The influence of further non-ventilation effects at each time-slice is taken up in the following section.

### 4.2 'Attenuation biases' in B-Atm offsets at the BA

As noted above, global average B-Atm changes can occur independently of ventilation effects, due to rapid atmospheric
radiocarbon variability that the deep ocean is too slow to respond to. We refer to such effects as 'attenuation biases', as they reflect the phase and attenuation response of a slowly adjusting reservoir (the global ocean interior), subject to continuous exchange with a more rapidly changing reservoir (the atmosphere) (Maier-Reimer and Hasselmann, 1987). The primary external (i.e. non-marine) driver for rapid atmospheric radiocarbon variability is likely to be changes in radiocarbon production (Köhler et al., 2022), though transient terrestrial carbon sources (e.g. from permafrost) might also be hypothesised (Köhler et
al., 2014; Wu et al., 2022).

For attenuation biases in mean ocean B-Atm offsets to occur, atmospheric radiocarbon activity would have to change rapidly, on a timescale that is shorter than the mean mixing time of the ocean (i.e. < 1000 yrs). Accordingly, gradual long-term trends in radiocarbon production are unlikely to produce significant attenuation biases, as is confirmed by model simulations where
atmospheric radiocarbon production is based on relatively smooth trends in mean relative (geomagnetic) palaeointensity (RPI) (Dinauer et al., 2020). However, given the scatter amongst existing reconstructions of past radiocarbon production (e.g. Laj et al., 2004; Adolphi et al., 2018; Nowaczyk et al., 2013; Channell et al., 2018), particularly on (sub-) millennial time scales, it remains unclear to what extent rapid (centennial/millennial) radiocarbon production variability, and/or other non-marine carbon sources, might have biased B-Atm offsets via their impact on the atmosphere. Therefore, to explore the maximum
possible contribution of externally driven atmospheric radiocarbon variability to mean ocean B-Atm changes, we compare transient simulations using the Bern3D model where radiocarbon production rates or atmospheric radiocarbon are held constant





(i.e. FIX- and CONST-, respectively) with simulations where nearly all variability in atmospheric radiocarbon is assumed to have occurred independently of ocean ventilation change (i.e. INT-, where atmospheric radiocarbon is prescribed according to *Intcal20*). The difference between the INT- simulations and their CONST/FIX- counterparts yields the *maximum* amplitude

of B-Atm changes that could be produced independently of ocean ventilation change, for each idealised scenario. Indeed, as discussed below, these estimates are likely over-estimates, as they are premised on the assumption that all atmospheric radiocarbon variability occurred independently of ocean-atmosphere radiocarbon exchange, which is unlikely.

As illustrated in **Figure 9** (and summarised in **Table 2**), our idealised scenarios demonstrate three key points regarding the

emergence of 'attenuation biases' in B-Atm offsets:

1) these biases depend on the occurrence of rapid atmospheric variability that is not forced by ocean ventilation change, but they can in principle result in persistent long term biases *via* an accumulation of centennial/millennial perturbations (e.g. as for a 'random walk' process);

2) such biases are time-varying, and may be positive relative to a given reference time (e.g. at the Laschamps event,

**Figure 9a**), negative (e.g. notably, at the BA onset, **Figure 9c**), or nil (e.g. at the LGM, **Figure 9b**), thus enhancing, diminishing or not affecting the B-Atm changes that are expressed between time periods (**Figure 9d**); and

3) the relative contribution of such biases will be diminished and potentially eliminated to the extent that they coincide with large abrupt ocean ventilation changes, though their magnitude depends mainly on the amplitude of atmospheric radiocarbon production changes (e.g. yielding a relatively invariant offset between the INT- and CONST/FIX-

simulations for any given time-slice, **Figure 9a-d**).

It is important to stress that the true magnitude of such attenuation biases cannot be determined without prior detailed knowledge of the history and magnitude of both ocean ventilation changes and non-marine carbon/radiocarbon inputs to the atmosphere. Nevertheless, two observations are worth underlining. The first is that maximum estimates of the attenuation

biases that could hypothetically affect deglacial B-Atm offsets would only result in relatively minor biases in the incremental B-Atm changes between time-slices (ranging from -227 to +25 [14]C-years). Even if maximal attenuation biases are hypothesised, the observed deglacial mean ocean B-Atm trends must include a significant ventilation contribution. Indeed, such changes are directly attested to by proxy evidence for ocean transport and sea-ice change (McManus et al., 2004; Schüpbach et al., 2018), as well as the spatially heterogeneous patterns in marine B-Atm offsets that we present here (e.g.

**Figures 4 and 7**).

A second key observation is that correcting for such biases would tend to diminish the apparent 'ventilation surge' that might be inferred from the change in B-Atm between HS and the BA (and more so between the LGM and the BA). Thus, if the rapid atmospheric radiocarbon decline across HS1 and at the onset of the BA was entirely driven by radiocarbon production changes

and/or non-marine carbon inputs to the atmosphere, then the change in global average B-Atm between the LGM and the BA





would be biased by at most ~ 227 $^{14}$C yrs. In this case, radiocarbon ventilation ages during the BA would be ~ 227 $^{14}$C yrs *older* than inferred from observed B-Atm offsets, resulting in a smaller radiocarbon ventilation change between the LGM and the BA. Non-ventilation biases affecting global average B-Atm differences between the LGM and the BA could therefore imply a smaller contribution from gas-exchange and transport rate changes to the apparent radiocarbon 'BA ventilation surge'

suggested in **Figure 6b, c**. In this case, the 'rejuvenation' of the marine radiocarbon pool may not in fact have been completed mid-way through deglaciation as initially apparent (Rafter et al., 2022). This inference would be more consistent with stable carbon isotope ($^{13}$C/$^{12}$C) evidence (Sikes et al., 2016b), which suggests an ongoing contribution to ocean-atmosphere carbon exchange well beyond the BA. Indeed, a significant portion of the convergence between marine and atmospheric radiocarbon observed at the BA may have been driven by 'old' carbon release to the atmosphere, e.g. from melting permafrost (Köhler et

al., 2014; Wu et al., 2022).

Although they remain difficult to accurately quantify, 'attenuation biases' unrelated to ventilation are important to acknowledge, as these may exert a subtle yet potentially significant influence on our interpretation of transient changes in B-Atm offsets and their quantification, as illustrated above for the BA. These effects will need to be explored in greater detail in

the future; a task that will inevitably require the use of numerical models, and that will also require accurate knowledge of past radiocarbon production changes (Köhler et al., 2022; Dinauer et al., 2020). The accuracy of our existing radiocarbon production records is discussed further in section 4.4.

### 4.3 Ventilation-related atmospheric CO$_2$ change

In theory, a broadly linear relationship between atmospheric CO$_2$ change and global average marine radiocarbon age anomalies is to be expected, when these are driven by gas-exchange and/or transport rates (Skinner and Bard, 2022; Skinner et al., 2017). This is because: 1) longer residence times in the ocean interior result in greater respired carbon accumulation along with greater radiocarbon decay; and 2) restricted gas exchange in regions of 'upwelling' and/or deep mixing impede the conversion of respired and/or disequilibrium carbon to equilibrium carbon (Eggleston and Galbraith, 2018), while also impeding the invasion

of radiocarbon into the ocean.

Our model sensitivity tests, involving shifts in vertical diffusivity, Southern Ocean winds, and/or gas exchange, cohere with a number of existing simulations using box-models and Earth System models of intermediate complexity (e.g. Tschumi et al., 2011; Kwon et al., 2011; Skinner and Bard, 2022), confirming a broad relationship between B-Atm anomalies and associated

atmospheric CO$_2$ change (**Table 3** and **Figure 10a**, symbols), despite large differences in model experiment set-up. Note that these experiments include the effects of pCO$_2$ changes on air-sea $^{14}$C exchange (Galbraith et al., 2015; Bard, 1988). Our sensitivity tests indicate consistent sensitivities for individual suites of experiments, e.g. for varying Southern Ocean winds, vertical diffusivity, or Southern Ocean gas exchange rates. However, depending on the processes responsible for altering deep ocean ventilation, modelled sensitivities span a range of approximately ±50% for a given magnitude of B-Atm change. It is



also worth noting that the sensitivities may vary with perturbation timescale, with different impacts on millennial timescales
than on glacial-interglacial or longer timescales (Jeltsch-Thömmes and Joos, 2020).

The broadly linear scaling apparent for each suite of model sensitivity tests is consistent with basic theory (Skinner et al., 2017;
Skinner and Bard, 2022), based on a two-box ocean connected to an atmosphere to form a closed system (**Figure 10a**, solid

line). This simple inventory theory predicts a higher sensitivity for higher global export productivity and/or a higher 'Revelle
buffer factor' (i.e. higher background atmospheric $pCO_2$), all else being equal. Clearly, the marine carbon cycle response to
the variety of ventilation processes that can affect radiocarbon cannot be reduced to a single linear scaling. However, the degree
of consistency in the parallel sensitivities of atmospheric $CO_2$ and marine B-Atm offsets implies that the wide range of
modelled sensitivities may be approximated by a theoretical prediction using an arbitrary ± 50 % range in export productivity

as a tuning parameter in the 2-box ocean model (**Figure 10a**, broken lines). This theoretical scaling, arbitrarily tuned to the
range of more complex model outputs, would suggest -6.3 ± 3.2 ppm $CO_2$ change per 100 $^{14}C$yrs of global mean B-Atm
change. Such a sensitivity would tentatively imply a drawdown of atmospheric $CO_2$ by ~ 53 ± 28 ppm associated with an
increase in global average B-Atm by ~808 ± 35 $^{14}C$ yrs, as reconstructed for our 'baseline' data flag scenario at the LGM
(**Table 1**), and assuming a maximum 'attenuation bias correction. Estimates based on uncorrected B-Atm offsets, and/or on

the two alternative data flagging scenarios ('low sedimentation' and 'Indian variant', **Table 1**), differ by only ~ 2 ppm. These
estimates should merely be interpreted as indicating a non-negligible contribution to deglacial $CO_2$ rise, perhaps equivalent to
over a third of the total glacial-interglacial atmospheric $CO_2$ change.

Interestingly, observed global average B-Atm estimates also correlate broadly with observed atmospheric $CO_2$ changes across

the last deglaciation, although the BA stands out as slightly anomalous (open symbols, **Figure 10b**). The correlation with
observed $pCO_2$ anomalies, particularly for the BA, is improved when global average B-Atm is 'corrected' for maximal possible
attenuation biases as discussed above (filled symbols in **Figure 10b**). The similarity of the observed correlation and the
modelled $CO_2$-sensitivity (red lines, **Figure 10b**) again merely suggests that a significant portion of the incremental changes
in atmospheric $CO_2$, stepping through the deglaciation from the LGM to the early Holocene, could in principle be accounted

for by ocean ventilation changes that influenced global average B-Atm offsets. Although the steeper dashed line in **Figure 10b**
would imply a maximal ~80 ppm contribution to deglacial atmospheric $CO_2$ rise due to ocean ventilation alone, we believe
this is unlikely. Rather, the observed correlation between atmospheric $CO_2$ and global average B-Atm offsets more likely
implies a mixture of direct *and* indirect causal connections that may have been coordinated by the thermal bipolar seesaw.
Direct impacts of ocean ventilation on atmospheric $CO_2$ (e.g. **Figure 10a**) would therefore have coincided with contributions

from linked processes, such as ocean temperature change, export productivity anomalies, etc. (Marchal et al., 1998; Menviel
et al., 2008; Jochum et al., 2022; Gottschalk et al., 2019; Menviel et al., 2012). In any event, if mean ocean B-Atm changes
scaled in a consistent manner with atmospheric $CO_2$ anomalies during deglaciation (as suggested by the relationships in **Figure
10**), then the ocean ventilation contribution to deglacial atmospheric $CO_2$ rise would have been primarily associated with HS1,



the BA and the YD. Indeed, the increases in atmospheric $CO_2$ observed after the YD and across the Holocene coincide with rather muted changes in global average B-Atm. This likely suggests a minor role for ocean 'ventilation' in $CO_2$ rise from the onset of the Holocene, which would be consistent with the interference that $CO_2$ rise from ~6 ka BP was primarily linked to changes in ocean temperature, the terrestrial biosphere, coral reef formation, and/or the solid Earth (i.e. volcanism, ocean alkalinity) (Joos et al., 2004; Broecker and Clark, 2007). This observation might also resonate with the speculative proposal of a 'natural tendency' for atmospheric decline across an interglacial due to ocean ventilation processes (Barker et al., 2019).

### 4.4 Towards a closure of the global carbon and radiocarbon cycles since the LGM

The reconciliation of past radiocarbon production changes with records of atmospheric $pCO_2$ and $\Delta^{14}C$ (hereafter $\Delta^{14}C_{atm}$) across the last deglaciation represents a long-standing puzzle that remains unresolved (Bard, 1998). This puzzle has a direct bearing on our understanding of past atmospheric $pCO_2$ change, as well as our understanding of geomagnetic and solar variability (Heaton et al., 2021). The record of atmospheric radiocarbon variability indicates a significant decrease across the last deglaciation, equivalent to a change in $\Delta^{14}C_{atm}$ of just over ~400 permil (Reimer et al., 2020) (**Figure 11b**, blue line). If there was no change in the steady state global radiocarbon inventory, the 90 ppm increase in atmospheric $pCO_2$ that occurred since the last glacial period would alone account for only ~25 permil of this change (Bard, 1998; Siegenthaler et al., 1980). This implies a dominant role for changes in the global radiocarbon budget (i.e. radiocarbon production), and/or the distribution of radiocarbon between atmosphere and other carbon reservoirs (i.e. the carbon cycle).

Model simulations of deglacial radiocarbon production and atmospheric radiocarbon and $pCO_2$ consistently indicate that past $\Delta^{14}C_{atm}$ cannot be accounted for by existing reconstructions of changing radiocarbon production alone (e.g. Hain et al., 2014; Kohler et al., 2006; Dinauer et al., 2020; Köhler et al., 2022). Earth System model simulations applying mean relative palaeomagnetic intensity (RPI) based radiocarbon production rate changes (Dinauer et al., 2020), yield only a small increase (~150 permil) in $\Delta^{14}C_{atm}$ at the LGM relative to the late Holocene (**Figure 11b**, black line). Similar $\Delta^{14}C_{atm}$ changes are obtained using alternative radiocarbon production histories (Hain et al., 2014; Kohler et al., 2006; Dinauer et al., 2020; Köhler et al., 2022). The widely recognised implication of these results is that additional carbon cycle changes (i.e. altered rates of carbon exchange with other reservoirs) and/or different radiocarbon production changes are required in order to account for the observed $\Delta^{14}C_{atm}$ amplitude.

Turning to carbon cycle changes first: given the tight coupling of the marine and atmospheric carbon pools, altered exchange rates between the ocean and atmosphere are likely to have played a leading role in deglacial $\Delta^{14}C_{atm}$ variability (Muscheler et al., 2004). Arguably for the first time, our global average B-Atm estimates confirm such a role. Indeed, our mean ocean B-Atm estimates indicate a lower average exchange rate of radiocarbon (and likely $CO_2$) between the ocean and atmosphere during the last glacial period, and an increase in this exchange rate across the deglaciation (**Figure 11c**, black line and circles). All else being equal, an increase in ocean-atmosphere radiocarbon exchange would result in a drop in $\Delta^{14}C_{atm}$ during





deglaciation, in parallel with a decrease in marine B-Atm offsets, as observed in **Figure 11b, c**. Such changes would have also contributed to deglacial atmospheric $CO_2$ rise (e.g. Muglia et al., 2018; Khatiwala et al., 2019; Brovkin et al., 2012; Ganopolski

and Brovkin, 2017). As discussed above, a tentative quantification of this contribution to atmospheric $CO_2$ rise can be derived from the observed mean ocean B-Atm (**Table 1**) and modelled sensitivities (**Figure 10**), as illustrated in **Figure 11d** (open circles and shaded region). This tentative contribution compares well with simulated $CO_2$ effects in the Bern3D model (INT_ALL60), bearing in mind that the simulated aging of the global ocean is only ~ 50% of that observed (**Figure 11c**, blue line and grey dashed line), and that ~50% of the simulated $CO_2$ signal derives from ocean ventilation impacts alone.


While the observed mean ocean B-Atm estimates suggest a significant impact on the carbon cycle, the observed changes are still too small to account for the observed ~400 permil drop in $\Delta^{14}C_{atm}$. This is demonstrated by model simulations that produce a mean ocean aging of ~ 500 $^{14}$C yrs (~63% of the observed value, **Table 1**), and yield a $\Delta^{14}C_{atm}$ increase of only ~56 permil, or ~14% of observed (**Figure 11b**, grey dashed line). Similar results have been obtained using the BICYCLE box-model

(Kohler et al., 2006), which produced a $\Delta^{14}C_{atm}$ increase of only ~200 permil (~50% of observed), despite yielding mean ocean B-Atm offsets that match our observed values (**Figure 11b,c**, dashed orange line). A mismatch was also obtained for the LGM using the CLIMBER intermediate complexity model (Ganopolski and Brovkin, 2017), where a 20% higher radiocarbon production rate, combined with reduced ocean ventilation causing a global mean B-Atm increase of ~800 years (again in line with our LGM estimate), coincided with a $\Delta^{14}C_{atm}$ increase of only ~280 permil (~70% of observed).


Therefore, existing radiocarbon production records cannot account for past atmospheric radiocarbon variability, either alone or in conjunction with ocean ventilation changes that are consistent with global mean B-Atm estimates (**Figure 11b, c**). While there remain uncertainties in atmospheric radiocarbon reconstructions, these are likely on the order of ~1% (Reimer et al., 2020), and it seems highly unlikely that $\Delta^{14}C_{atm}$ variability over the last glacial cycle has been significantly overestimated.

Similarly, it seems implausible (though clearly not impossible) that existing marine radiocarbon data significantly underestimate the magnitude of mean ocean B-Atm change since the last glacial period. The problem of closing the global radiocarbon budget since the last glacial becomes even more difficult if volcanic/metamorphic $CO_2$ inputs are invoked as a significant contributor to the global carbon pool during the last glacial period (Stott et al., 2019; Stott et al., 2009).

Given the wide range of existing radiocarbon production estimates (e.g. as compiled by Dinauer et al., 2020) (**Figure 11a**, shaded area), it seems reasonable to postulate that existing production reconstructions might, on average, underestimate the amplitude of radiocarbon production rate change between the last glacial and the late Holocene, as suggested by a recent box-model study (Köhler et al., 2022). A 'polar bias' in radiocarbon production records derived from ice-core $^{10}$Be fluxes has indeed been recently quantified separately for the geomagnetic and heliomagnetic modulations of cosmogenic production

(Adolphi et al., 2023). Nevertheless, it has also been noted that correcting for the identified long-term geomagnetic bias in $^{10}$Be the would not eliminate the mismatch between observed and modelled $\Delta^{14}C_{atm}$ at the LGM (Adolphi et al., 2023). Our



idealised simulations with prescribed $\Delta^{14}C_{atm}$ allow us to infer the radiocarbon production changes that would be needed to reconcile imposed ventilation/carbon cycle changes with observed atmospheric $\Delta^{14}C_{atm}$ (**Figure 10a**, blue line). The rapid fluctuations in production that are inferred across the last deglaciation (e.g. across HS1, the BA and YD) almost certainly

reflect biases due to transient carbon cycle and ocean ventilation changes that have not been implemented in the idealised simulations (see section 2.5) However, the longer-term trend in inferred production rates indicates levels at the LGM that are close to the high end of the existing range of estimates (**Figure 11a**, shaded area). A similar result has recently been obtained using the BICYCLE box-model (Köhler et al., 2022). The higher radiocarbon production rates that are inferred at the LGM would have had a significant impact, possibly accounting for the bulk of the deglacial atmospheric radiocarbon signal (**Figure**

**11b**, dashed blue line). The implication of these results is that a parallel closure of the radiocarbon and carbon cycles since the last glacial period might yet be obtained by exploiting the plausible range of reconstructed radiocarbon production rates (Köhler et al., 2022). Our global average B-Atm estimates provide a useful new constraint for achieving this goal.

## 5 Conclusions

We present spatial interpolations of compiled radiocarbon data for a suite of time-slices spanning the last deglaciation. The primary purpose of these interpolations is to derive global average B-Atm estimates. A clear trend in global average B-Atm offsets is apparent from the LGM (when B-Atm was ~800 $^{14}$C yrs higher than modern on average), demonstrating unambiguous changes in the partitioning of radiocarbon between the ocean and atmosphere since the last deglaciation.

The spatial interpolations cohere with previous studies in indicating a stepwise and spatially heterogenous rejuvenation of the ocean interior across the last deglaciation, and in suggesting the operation of a 'ventilation seesaw' between the North Atlantic and the North Pacific/Southern Ocean, especially during HS1, the YD, and the EHOL.

A comparison of surface-, shallow- and deep-water B-Atm trends indicates that transport changes in the upper ocean across

the last deglaciation were likely modest (Marchal and Zhao, 2021a), and that B-Atm changes in upper ocean (< 2 km) were more strongly influenced by evolving gas-exchange efficiency at high latitudes. In contrast, a more significant contribution from evolving transport and/or water mass geometry is apparent in the deeper ocean, > 2km.

The time-slice reconstructions emphasize a widespread drop in B-Atm at the onset of the BA, resulting in a global average B-

Atm within ~100 $^{14}$C yrs of modern. However, model sensitivity tests indicate that a portion of this B-Atm drop may have resulted from atmospheric radiocarbon dynamics that were independent of ocean ventilation (e.g. radiocarbon production, terrestrial carbon release, etc.). The exact magnitude of this effect cannot yet be quantified, but a maximum bias of ~ +190 $^{14}$C yrs relative to the LGM is estimated. Such a bias would imply that mean ocean B-Atm at the BA underestimates the true 'ventilation age' of the ocean.


Model sensitivity tests further suggest a direct relationship between global average B-Atm anomalies and atmospheric $CO_2$ change, with a tentative average sensitivity of ~ -6.3 ppm $CO_2$ per 100 $^{14}$C yrs. On this basis, our global average B-Atm estimates would imply a non-negligible contribution to atmospheric $CO_2$ change across the last deglaciation (perhaps as much as >30% of the total observed). Global average B-Atm estimates also suggest that any ventilation contribution to atmospheric

$CO_2$ change was concentrated during HS1, the BA and the YD, and was largely exhausted by the onset of the Holocene.

While our results serve to underline, and tentatively to quantify, the ocean's role in deglacial carbon cycle change, they also demonstrate that a complete closure and reconciliation of the radiocarbon and carbon cycles since the last glacial remains to be achieved. Our results point to the possibility that, on average, existing reconstructions may tend to underestimate

radiocarbon production rates during the last glacial period. Further work to improve the accuracy of past radiocarbon production rate estimates therefore emerges as a priority.

**Data availability**

Data presented in this study are lodged with the PANGAEA database at: https://www.pangaea.de.


**Author contributions**

LCS designed the study, compiled and processed the radiocarbon data, and performed the interpolations with the assistance of FP. FP developed the interpolation code, and LCS and FP analysed the interpolation outputs. Numerical model runs using Bern3D were performed by AJ-T and FJ, and analysed by AJ-T, FJ and LCS. LCS wrote the manuscript with input from all

co-authors.

**Competing interests**

The authors declare that there are no competing financial and/or non-financial interests in relation to the work described.

**Acknowledgements**

This work benefited from discussions during the INQUA IPODS working group meeting held Cambridge in 2018. LCS acknowledges support from NERC grant NE/L006421/1, the Royal Society and the Cambridge Isaac Newton Trust. AJ-T and FJ acknowledge funding from the Swiss National Science Foundation (SNF 200020_200511). The authors thank Laurie Menviel for her assistance accessing the LOVECLIM results used to test the interpolation method, as well as Patrick Rafter,

Ning Zhao, Dan Amrhein, and Olivier Marchal for helpful discussions. This study was initiated in 2019 and initially submitted for review in 2021; we are grateful for the constructive comments of one anonymous reviewer received at that stage, which helped to improve an earlier draft of the manuscript.

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



**Table 1. Time-slice global average B-Atm values and anomalies *versus* modern, based on 3D interpolations, for three data flagging approaches, as described in the main text: 'baseline' (all flagged data are excluded); 'low sedimentation' (as for baseline, but with low sedimentation rate sites included); and 'Indian variant' (as for baseline, but with data from a single site in the deep Indian Ocean also omitted. Hypothetical corrections for maximum 'attenuation biases' and inferred atmospheric CO₂ impacts are shown compared with observed mean atmospheric CO₂ levels.**

| Time slice | Baseline | | | Low sedimentation | | | Indian variant | | | Max. attenuation bias in model (CONST) | "Corrected" D(B-Atm) | Estimated pCO2 impact | Obs pCO2 | Obs D(pCO2) vs PI | err |
|---|---|---|---|---|---|---|---|---|---|---|---|---|---|---|---|
| | Mean ocean B-Atm | D(B-Atm) vs PI | err | Mean ocean B-Atm | D(B-Atm) vs PI | err | Mean ocean B-Atm | D(B-Atm) vs PI | err | | | | | | |
| | 14C yrs | 14C yrs | 14C yrs | 14C yrs | 14C yrs | 14C yrs | 14C yrs | 14C yrs | 14C yrs | 14C yrs | 14C yrs | ppmv | ppmv | ppmv | ppmv |
| HOL | 1334 | -25 | 22 | 1319 | -40 | 25 | 1334 | -25 | 22 | 25 | -50 | 3 | 281.9 | -3.6 | 0.5 |
| EHOL | 1357 | -2 | 26 | 1361 | 2 | 39 | 1360 | 1 | 25 | -52 | 50 | -3 | 268.2 | -17.3 | 0.4 |
| YD | 1485 | 126 | 35 | 1523 | 164 | 37 | 1485 | 126 | 35 | 28 | 98 | -6 | 253.7 | -31.8 | 0.7 |
| BA | 1430 | 71 | 31 | 1495 | 136 | 38 | 1430 | 71 | 31 | -217 | 288 | -18 | 241.5 | -43.9 | 0.5 |
| HS! | 1965 | 606 | 57 | 2022 | 663 | 57 | 1878 | 519 | 40 | -107 | 713 | -45 | 219.7 | -65.8 | 1.5 |
| LGM | 2167 | 808 | 35 | 2192 | 833 | 37 | 2142 | 783 | 34 | -26 | 834 | -53 | 194.7 | -90.8 | 0.6 |





**Table 2. B-Atm and atmospheric CO₂ anomalies obtained for sensitivity experiments, evaluated at a series of time intervals, using the Bern3D model under varying Southern Ocean wind, Southern Ocean gas-exchange, vertical diffusivity, or all combined (as illustrated in Figure 9 of the main text).**

| Scenario | Ventilation parameter changed | Reduction (%) | INT D(B-Atm) vs PI | INT D(pCO2) vs PI | FIX D(B-Atm) vs PI | FIX D(pCO2) vs PI | CONST D(B-Atm) vs PI | CONST D(pCO2) vs PI |
|---|---|---|---|---|---|---|---|---|
| **41 ka** | Control | 0 | 1165 | -13 | 92 | -13 | 100 | -13 |
| | S.O. wind | 20 | 1220 | -17 | 128 | -17 | | |
| | | 40 | 1239 | -20 | 139 | -20 | | |
| | | 60 | 1215 | -21 | 120 | -21 | | |
| | | 80 | 1151 | -20 | 77 | -20 | | |
| | S.O gas exchange | 20 | 1223 | -14 | 132 | -14 | | |
| | | 40 | 1291 | -15 | 179 | -15 | | |
| | | 60 | 1370 | -17 | 233 | -17 | | |
| | | 80 | 1465 | -21 | 300 | -21 | | |
| | vertical diffusivity | 20 | 1237 | -16 | 142 | -16 | | |
| | | 40 | 1323 | -19 | 202 | -19 | | |
| | | 60 | 1401 | -21 | 258 | -21 | | |
| | | 80 | 1466 | -23 | 305 | -23 | | |
| | all | 20 | 1363 | -21 | 227 | -21 | | |
| | | 40 | 1524 | -28 | 339 | -28 | | |
| | | 60 | 1628 | -32 | 411 | -32 | 418 | -32 |
| | | 80 | 1627 | -30 | 408 | -30 | | |
| **LGM** | Control | 0 | 219 | -19 | 143 | -19 | 148 | -19 |
| | S.O. wind | 20 | 248 | -22 | 176 | -22 | | |
| | | 40 | 252 | -25 | 180 | -25 | | |
| | | 60 | 227 | -26 | 152 | -25 | | |
| | | 80 | 183 | -24 | 105 | -24 | | |
| | S.O gas exchange | 20 | 254 | -20 | 182 | -20 | | |
| | | 40 | 295 | -21 | 227 | -21 | | |
| | | 60 | 342 | -23 | 280 | -23 | | |
| | | 80 | 400 | -26 | 344 | -26 | | |
| | vertical diffusivity | 20 | 269 | -21 | 199 | -21 | | |
| | | 40 | 328 | -23 | 264 | -23 | | |
| | | 60 | 384 | -25 | 325 | -25 | | |
| | | 80 | 417 | -26 | 376 | -26 | | |
| | all | 20 | 338 | -26 | 275 | -26 | | |
| | | 40 | 467 | -33 | 418 | -33 | | |
| | | 60 | 520 | -36 | 477 | -36 | 477 | -36 |
| | | 80 | 476 | -34 | 429 | -34 | | |
| **BA** | Control | 0 | -96 | -10 | 120 | -10 | 109 | -10 |
| | S.O. wind | 20 | -96 | -9 | 119 | -9 | | |
| | | 40 | -104 | -9 | 109 | -9 | | |
| | | 60 | -120 | -8 | 90 | -8 | | |
| | | 80 | -144 | -7 | 61 | -7 | | |
| | S.O gas exchange | 20 | -87 | -10 | 129 | -10 | | |
| | | 40 | -77 | -10 | 140 | -10 | | |
| | | 60 | -66 | -10 | 152 | -10 | | |
| | | 80 | -54 | -10 | 166 | -10 | | |
| | vertical diffusivity | 20 | -77 | -10 | 141 | -10 | | |
| | | 40 | -59 | -11 | 163 | -11 | | |
| | | 60 | -38 | -11 | 186 | -11 | | |
| | | 80 | -20 | -11 | 206 | -11 | | |
| | all | 20 | -65 | -10 | 154 | -10 | | |
| | | 40 | -44 | -10 | 177 | -10 | | |
| | | 60 | -34 | -9 | 186 | -9 | 101 | -9 |
| | | 80 | -41 | -9 | 178 | -9 | | |




*Table 3. B-Atm and atmospheric CO₂ results for sensitivity experiments carried out using the Bern3D model with varying Southern Ocean wind, gas exchange, or global vertical diffusivity (illustrated in Figure 10(a) of the main text). Results are evaluated 2000 years after forcing is applied (see text).*

| Scenario | Ventilation parameter changed | Reduction (%) | B-Atm | pCO2 | D(B-Atm) | D(pCO2) |
|---|---|---|---|---|---|---|
| PI (ctrl) | none | 0 | 1428 | 276.5857 | 0 | 0 |
| PI | S.O. wind | 20 | 1495 | 270.2083 | 67 | -6 |
| | | 40 | 1548 | 264.4817 | 120 | -12 |
| | | 60 | 1577 | 260.3277 | 150 | -16 |
| | | 80 | 1588 | 259.9954 | 160 | -17 |
| PI | S.O gas exchange | 20 | 1472.1 | 275.7578 | 44 | -1 |
| | | 40 | 1523.3 | 274.6609 | 96 | -2 |
| | | 60 | 1585.5 | 272.8658 | 158 | -4 |
| | | 80 | 1662.4 | 269.6315 | 235 | -7 |
| PI | vertical diffusivity | 20 | 1480.3 | 273.7565 | 53 | -3 |
| | | 40 | 1523.6 | 271.5337 | 96 | -5 |
| | | 60 | 1564.6 | 269.6152 | 137 | -7 |
| | | 80 | 1597.5 | 268.211 | 170 | -8 |







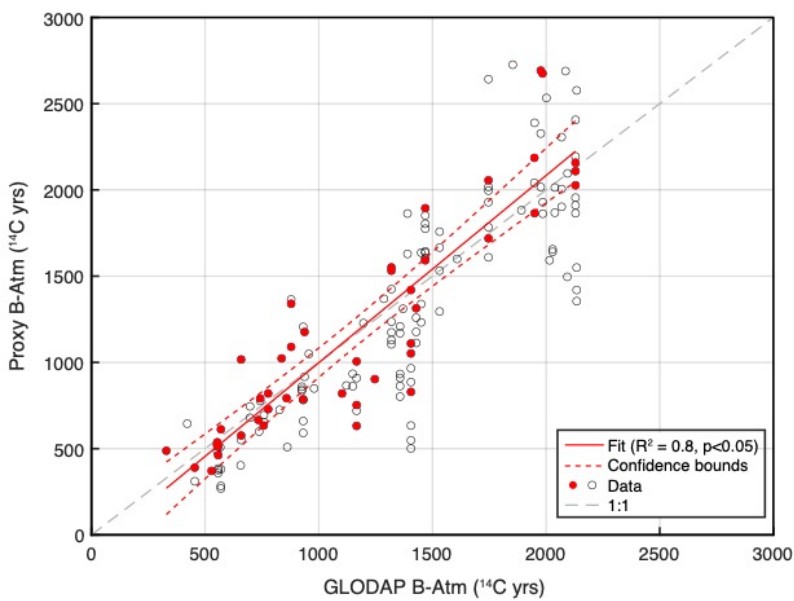

**Figure 1. Comparison of modern seawater (bomb-corrected, or 'background') B-Atm radiocarbon age offsets (Key et al., 2004) versus proxy-based B-Atm reconstructions using material deposited during the last ~6,000 years (black open circles) and the last ~1,500 years (red filled circles). Dashed grey line indicates the 1:1 trend. The linear fit to the data is indicated by the red line (dashed red lines show 95% confidence limits), with equation: $y = (1.1 \pm 0.1)x + (89 \pm 98)$, $R^2 = 0.81$, $p = 1.77 \times 10^{-17}$.**




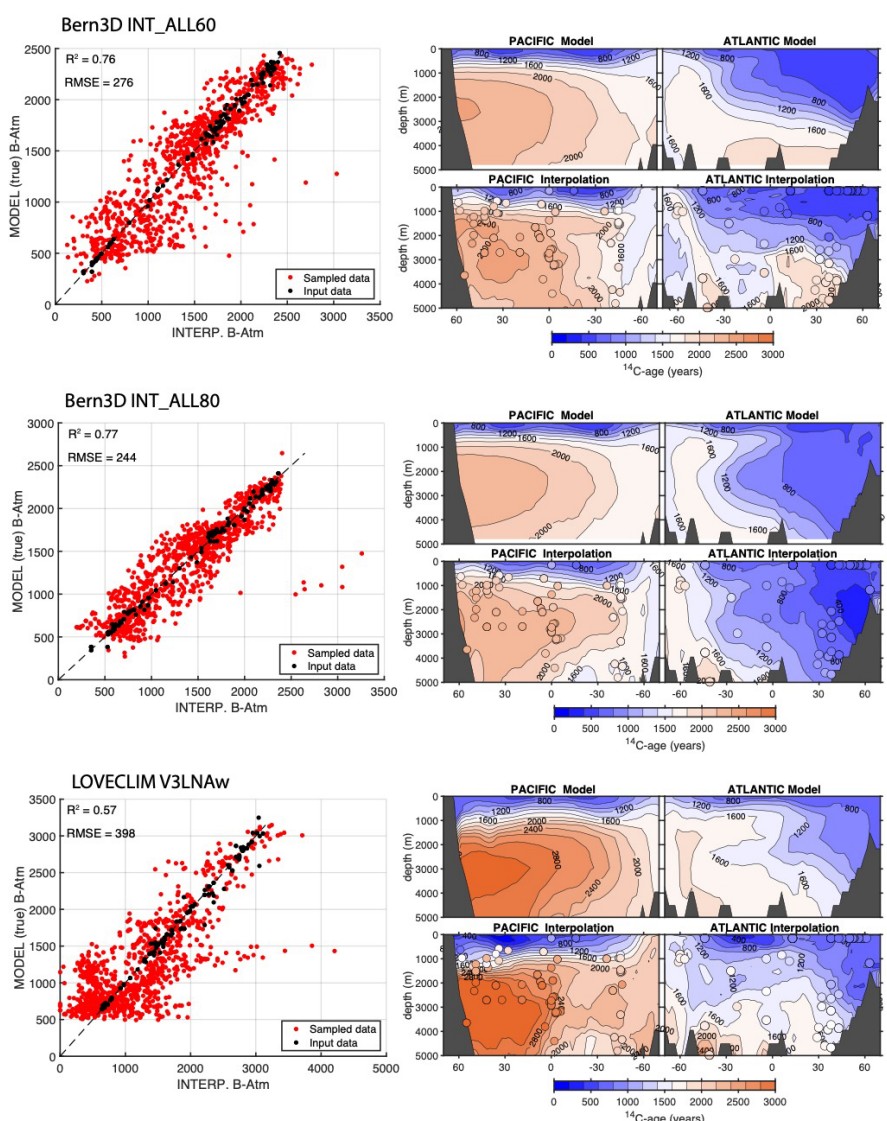

**Figure 2. Testing the interpolation method using numerical model outputs. Interpolations for the Bern3D and LOVECLIM models, from top to bottom: INT_ALL60 (this study), INT_ALL80 (this study), and V3LNAw (Menviel et al., 2017). Cross plots at left show 'true' versus interpolated B-Atm offsets (black circles for input data, red circles for 1000 randomly sampled locations in the ocean interior), with $R^2$ and RMSE shown. Contoured panels show zonally averaged B-Atm offsets (i.e. $^{14}$C-age relative to the atmosphere), for the Pacific (left) and Atlantic (right), for both the model output (upper panels) and the interpolated reconstructions (lower panels), with input data used for the interpolations indicated by the filled circles.**

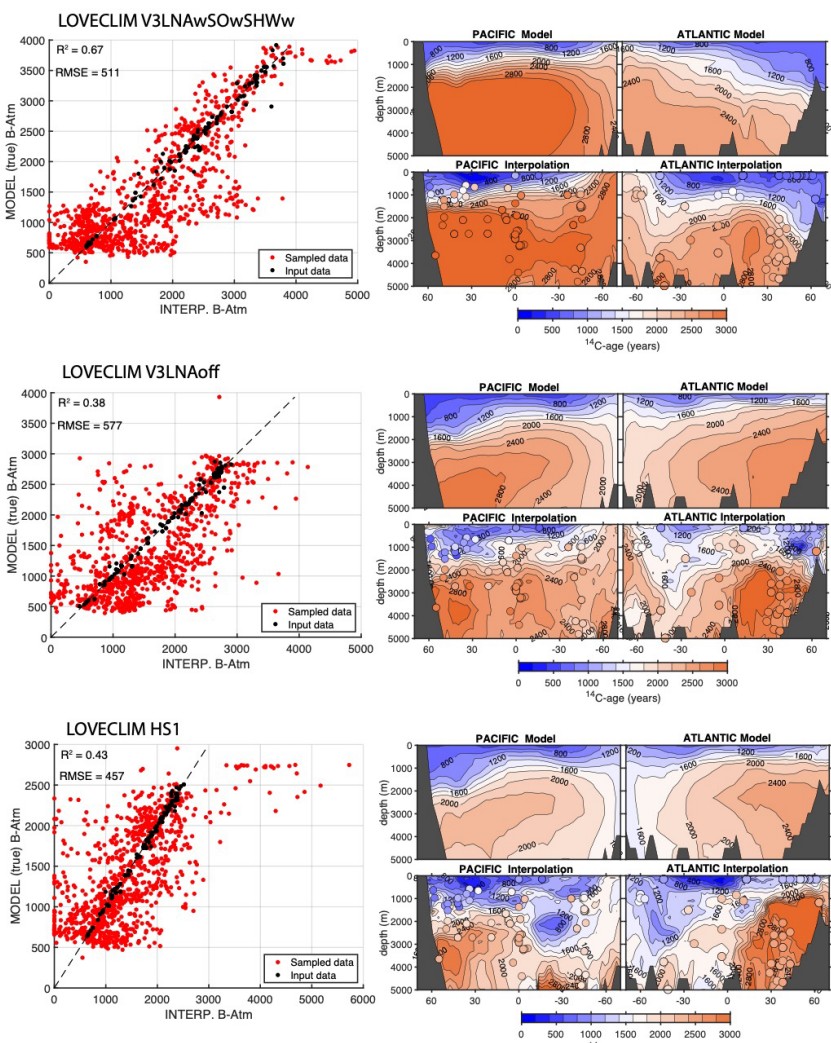

**Figure 3. Testing the interpolation method using numerical model outputs, as for Figure 2. Interpolations for simulations using the LOVECLIM model, from top to bottom: V3LNAwSOwSHWw (Menviel et al., 2017), V3LNAoff (Menviel et al., 2017), and HS1 (Menviel et al., 2018). Cross plots at left show 'true' versus interpolated B-Atm offsets (black circles for input data, red circles for 1000 randomly sampled locations in the ocean interior), with $R^2$ and RMSE shown. Contoured panels show zonally averaged B-Atm offsets (i.e. $^{14}$C-age relative to the atmosphere), for the Pacific (left) and Atlantic (right), for both the model output (upper panels) and the interpolated reconstructions (lower panels), with input data used for the interpolations indicated by the filled circles.**

1125

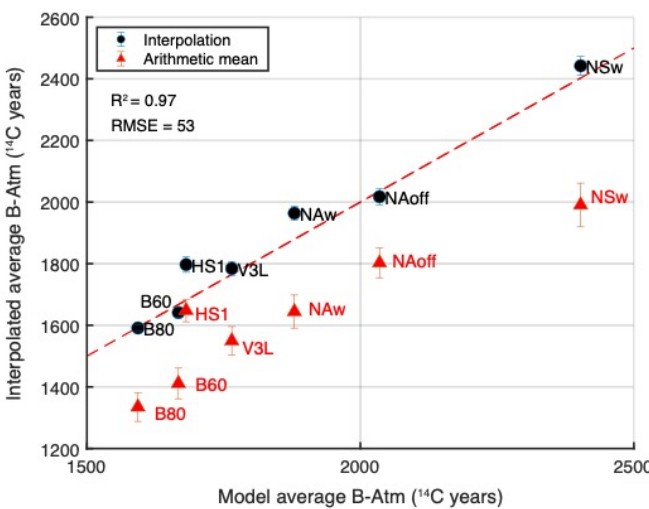

**Figure 4. Comparison of global average B-Atm offsets based on interpolated fields (circles, with estimated interpolation error), and geometric averages of the input data used for the interpolations (triangles, with standard error), versus the true values for the model simulations shown in Figure 1 and an additional simulation (NAw) from (Menviel et al., 2017) (V3LNAw). The $R^2$ correlation coefficient, RMSE, and 1:1 line (red dashed line) are indicated.**

1130

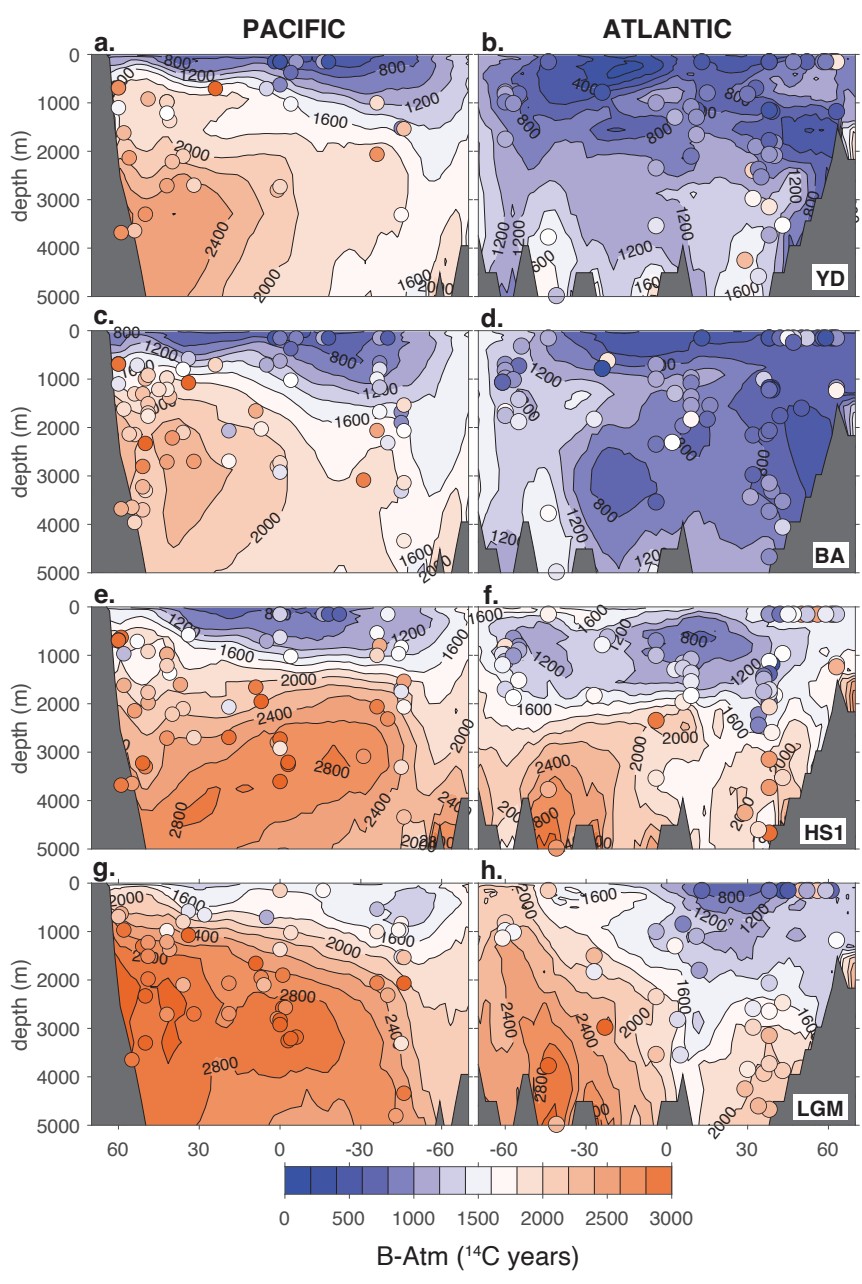

135

**Figure 5. Zonally averaged interpolated B-Atm radiocarbon age offsets for the LGM, HS1, BA, and YD (Pacific zonal averages at left, Atlantic at right). Filled circles and shading indicate input data and values**.



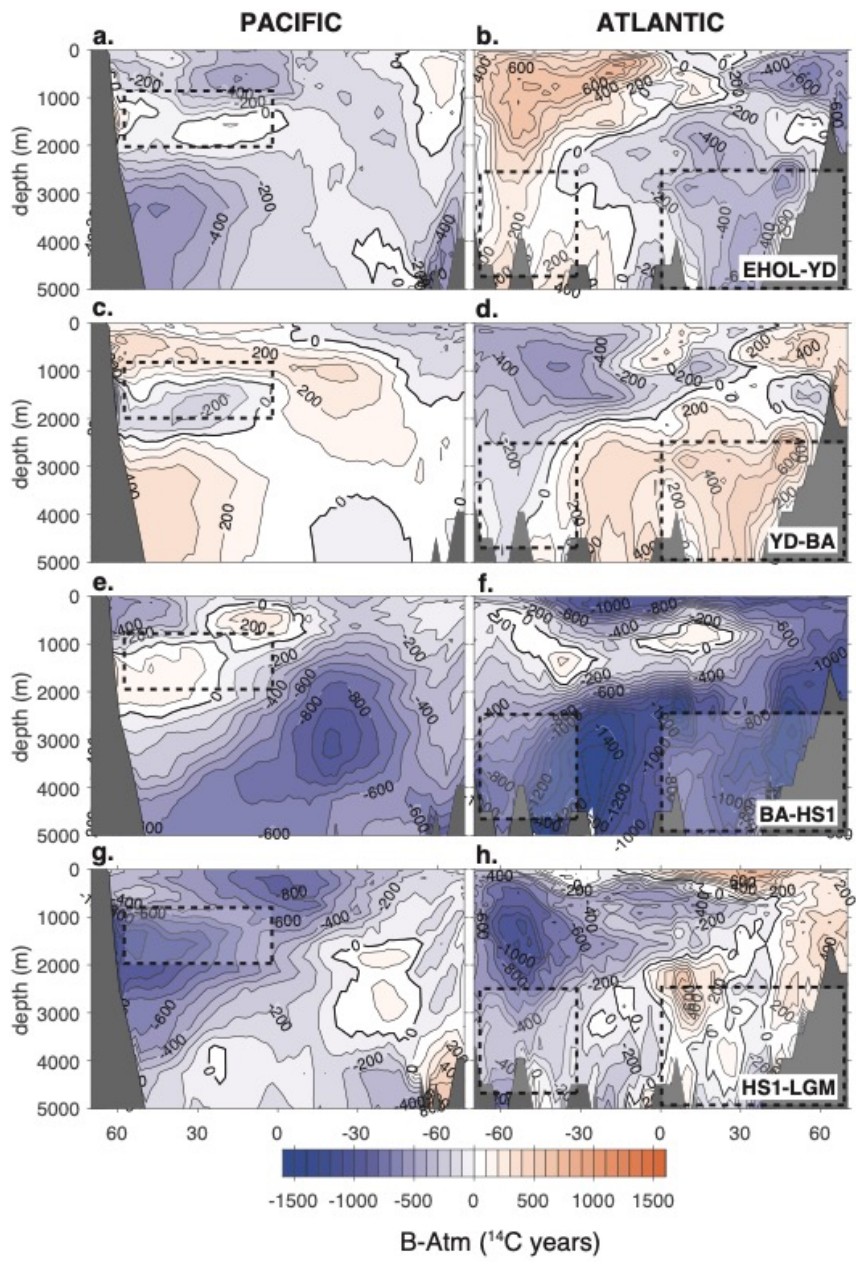

**Figure 6. Offsets between spatial B-Atm interpolations for successive time-slice reconstructions. Boxed areas highlight regions of the deep North Atlantic, deep Southern Ocean, and intermediate North Pacific, for which regional time-series splines are illustrated in Figure 8. Broadly antiphased anomalies are indicated between the North Atlantic and Southern Ocean (especially the Atlantic sector), and between the North Atlantic and intermediate North Pacific.**



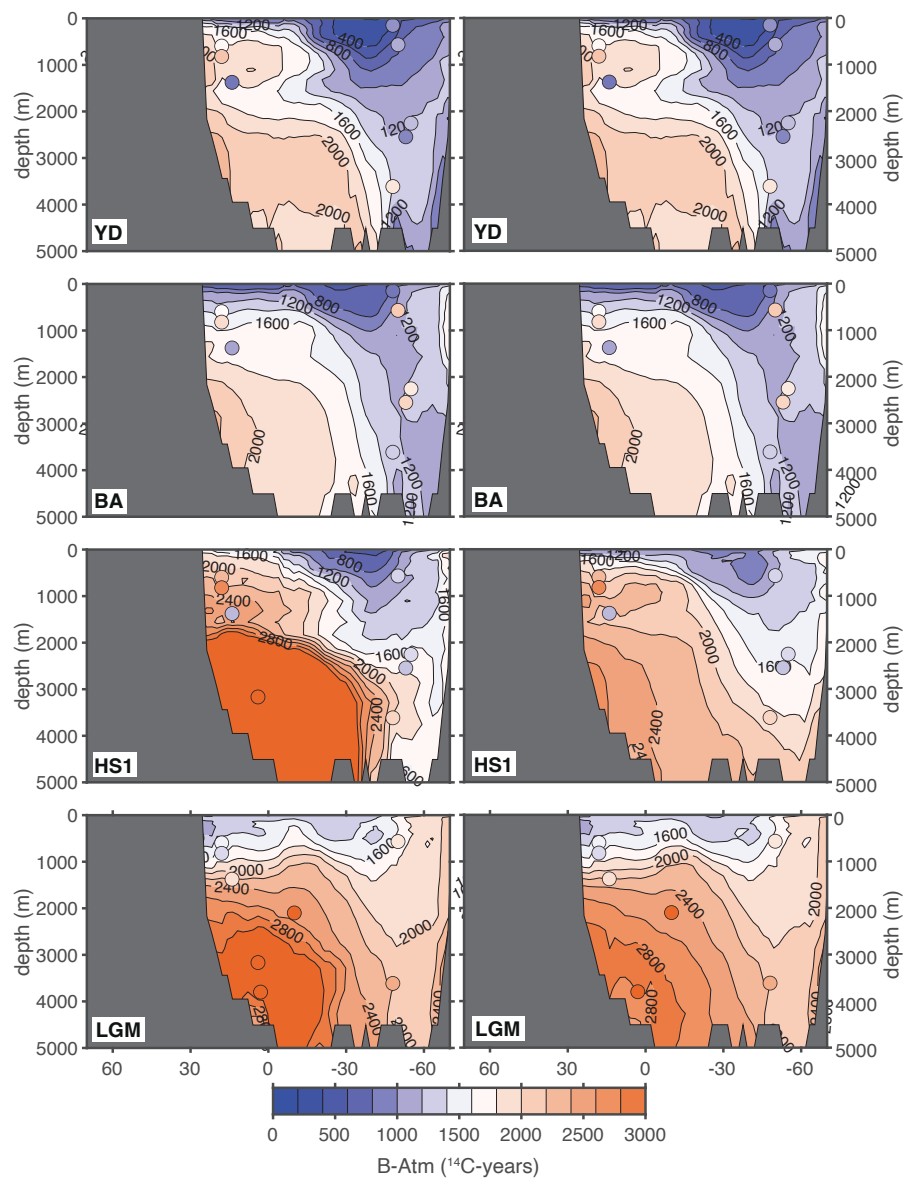

**Figure 7. Zonally averaged interpolated B-Atm radiocarbon age offsets in the Indian Ocean, for the LGM, HS1, BA, and YD. Sparse data coverage is notable. Left: time-slice reconstructions for the 'baseline' scenario including the data of Bharti et al. (2022), from the LGM and HS1, and with sedimentation rates just above 2cm/kyr. Right: time-slice reconstructions for the 'Indian variant' scenario, omitting the data of Bharti et al. (2022), indicating a significant impact on the interpolated field despite little impact on global mean values (Table 1).**

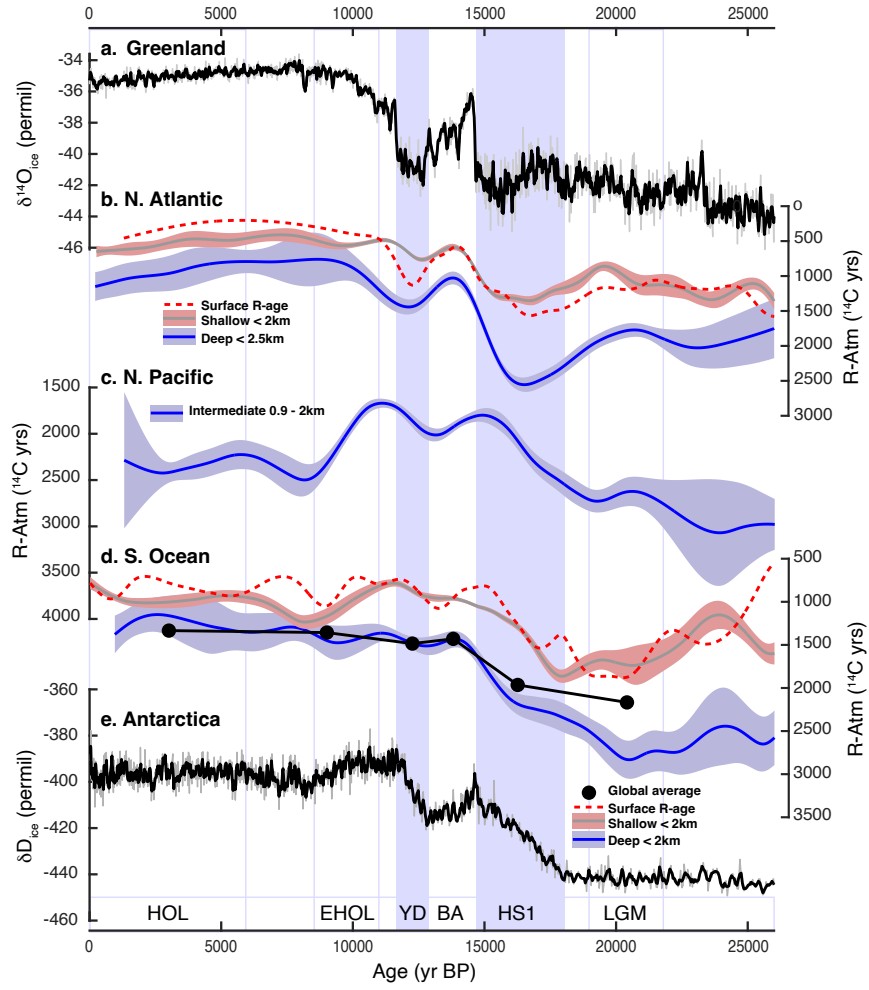

**Figure 8.** 'Ventilation seesaws' across the last deglaciation, based on cubic spline fits to compiled ocean-atmosphere radiocarbon age offsets (Skinner and Bard, 2022). (a) Greenland temperature proxy (Svensson et al., 2008). (b) NE Atlantic shallow sub-surface reservoir ages ((Skinner et al., 2019), dashed red line); B-Atm from the North Atlantic <2km (grey line, red shaded area); B-Atm from the deep North Atlantic >2km (blue line, blue shaded area). (c) B-Atm from the intermediate North Pacific (blue line, blue shaded area). (d) Mean ocean B-Atm estimates (black line and circles, Table 1), compiled shallow sub-surface reservoir ages from the Southern Ocean (Skinner et al., 2019) (dashed red line); B-Atm from the Southern Ocean <2km (grey line and red shaded area); B-Atm from the deep Southern Ocean >2km (blue line and blue shaded area). (e) Antarctic temperature proxy (Epica Community Members, 2004; Lemieux-Dudon et al., 2010). Vertical lines and shaded bars indicate the timing of the LGM, HS1, BA, YD EHOL and HOL time-slice.





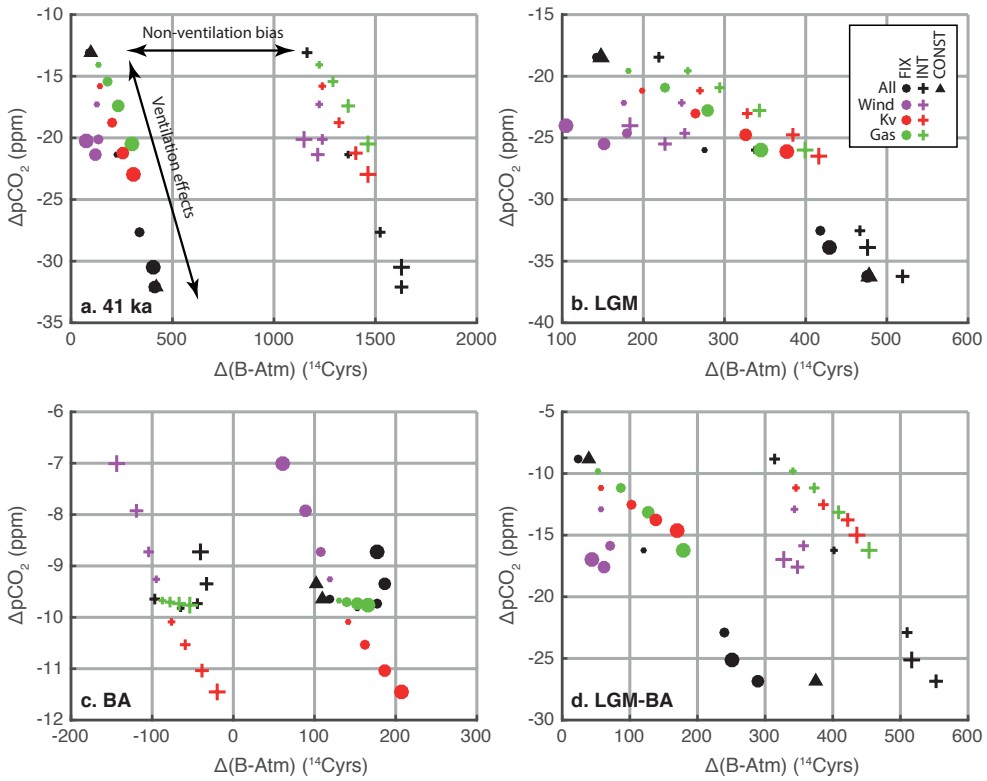

**Figure 9. Model outputs illustrating broadly consistent ocean ventilation effects on B-Atm offsets (i.e. linear trends for similar symbol types; sloping arrow in panel (a)), as well as evolving hypothetical 'attenuation biases' that are unrelated to ocean 'ventilation' indicated by horizontal offsets between similar symbol colours (horizontal arrow in panel (a)). (a) 41 ka BP (coinciding with the Laschamps geomagnetic excursion); (b) the LGM; (c) the BA; (d) the offset between the LGM and the BA. Data are expressed as 1000-yr average anomalies relative to the pre-industrial period 4-5 ka BP, when atmospheric radiocarbon was relatively stable and therefore plausibly approached a pseudo-equilibrium state. In all panels, crosses indicate model experiments carried out with prescribed atmospheric radiocarbon as given by Intcal20 (Reimer et al., 2020) (INT); circles are for atmospheric radiocarbon fixed at 140 permil (FIX); and triangles indicate are for constant pre-industrial radiocarbon production rates (CONST). Symbol colours distinguish experiments carried out with altered Southern Ocean winds (purple), vertical diffusivity (red), Southern Ocean gas-exchange efficiency (green), or all combined (black). Four symbol sizes indicate the extent of parameter'reduction: 0% (control), 20%, 40%, 60% and 80%. For CONST, only two sets of experiments were run: for control conditions, and for all tuning variables reduced by 60%.**

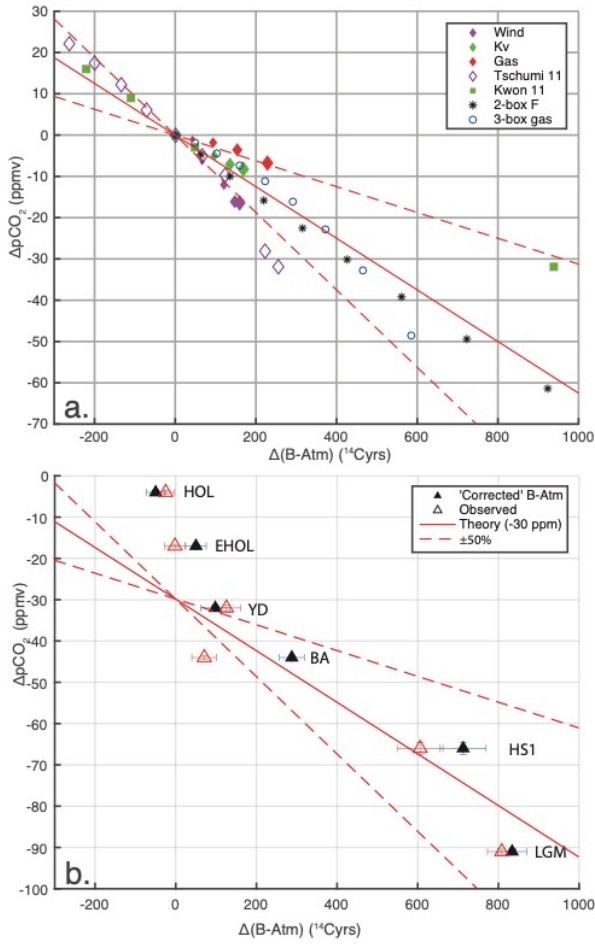

l180

**Figure 10. (a) Theoretical and modelled sensitivity of atmospheric CO₂ anomalies to marine radiocarbon ventilation age (B-Atm) anomalies, arising from air-sea gas exchange and transport changes: filled diamonds indicate model sensitivity tests from this study, based on step-changes under PI conditions and 2,000 years of equilibration time**

l185 **(purple, Southern Ocean wind; green, vertical diffusivity; red, Southern Ocean gas-exchange efficiency); open diamonds are Southern Ocean wind experiments (Tschumi et al., 2011); filled squares are wind and/or diffusivity experiments using an idealised radiocarbon-like tracer (Kwon et al., 2011); asterisks and open circles represent 2- and 3-box model experiments, with varying overturning rates (F) and 'high latitude' gas-exchange rates (gas), respectively. A median theoretical sensitivity is indicated by the solid and broken red lines, derived for the 2-box model results**

l190 **(Skinner and Bard, 2022), equivalent to -6.3 ± 3.2 ppm CO₂ change per 100 ¹⁴Cyrs of global radiocarbon ventilation age change. (b) Observed values for comparison, expressed as time-slice anomalies *versus* pre-industrial values: open triangles show paired observations of mean ocean B-Atm (this study) and atmospheric CO₂ (Monnin et al., 2001;**



**Marcott et al., 2014),; filled triangles show with observed B-Atm age offsets corrected for hypothetical 'non-ventilation'**

**biases (see text). Solid and broken red lines in panel b show the same theoretical sensitivities as for panel (a), offset by**

1195 **-30 ppm to aid comparison with observed trends.**

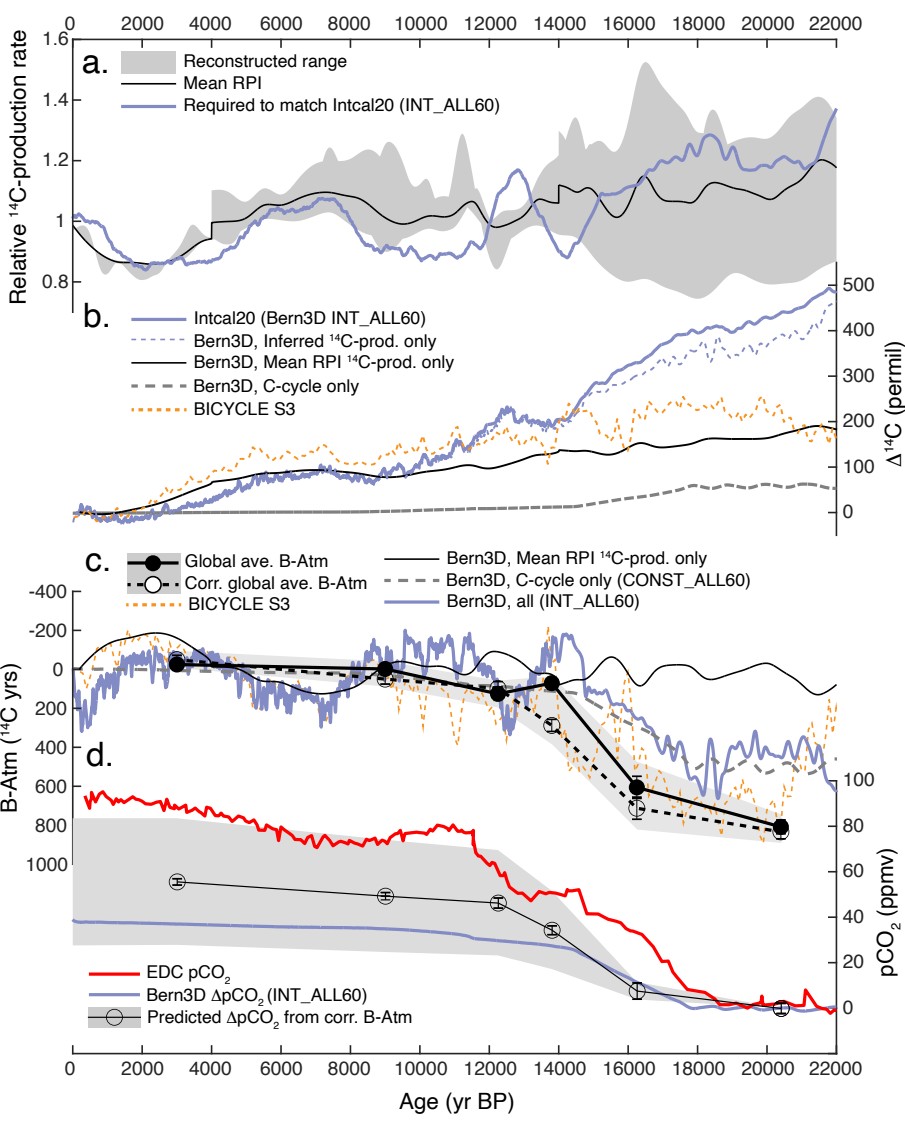

**Figure 11. Observed and modelled atmospheric and marine radiocarbon, compared with radiocarbon production rates, and atmospheric CO₂. (a) Relative radiocarbon production rates: shaded area, full range of reconstructed values (Laj et al., 2000; Laj et al., 2004; Adolphi et al., 2018; Channell et al., 2018; Nowaczyk et al., 2013); solid black line, mean RPI (Laj et al., 2000; Laj et al., 2004; Channell et al., 2018; Nowaczyk et al., 2013); heavy blue line, as inferred from idealised model scenario to match Intcal20, INT_ALL60 (1kyr moving average, this study). (b) Atmospheric Δ¹⁴C anomalies, normalised to modern: prescribed in the INT_ALL60 model scenario (i.e. Intcal20 (Reimer et al., 2020); solid dark blue line, this study); driven only by inferred production rates that match Intcal20 (INT_ALL60-CONST_ALL60; dashed blue line, this study); driven only by mean RPI production rates (Dinauer et al., 2020) (solid**





black line); driven only by carbon cycle/ventilation changes in the CONST-ALL60 model scenario (dashed grey line, this study); simulated with the BICYCLE box-model (Kohler et al., 2006) using [10]Be-based (Muscheler et al., 2005) radiocarbon production estimates and a full carbon cycle scenario (dashed orange line). (c) B-Atm radiocarbon age offset anomalies relative to modern: Bern3D and BICYCLE model outputs as for panel (b); and inferred from time-slice interpolations (black filled circles and line; this study), including a maximal correction for 'attenuation biases' (open black circles and dashed line), with the full range of reconstructed values (shaded area) due to uncertainties, corrections, and alternative data flagging scenarios (Table 1). (d) Atmospheric $CO_2$ normalised to LGM values: from EDC (Monnin et al., 2001) (red line); inferred from observed mean 'corrected' ocean B-Atm using a sensitivity of ~ -6.3 ± 3.2 ppm per 100 [14]Cyrs (open black circles and line with shaded range, this study); and simulated in the Bern3D model for the INT_ALL60 scenario that is also illustrated in panels (a)-(c).