# Peer review of "Rejuvenating the ocean: mean ocean radiocarbon, CO2 release, and radiocarbon budget closure across the last deglaciation"

_Climate of the Past, 2023_

## Author Comment (AC1)

**Reviewer 1:**

We are grateful for the helpful comments of Reviewer 1, which have helped us to improve the manuscript. We provide our responses below, embedded in the original commentary.

Skinner et al., present a compilation of ocean-atmosphere radiocarbon age offset (B-Atm) across the last deglaciation. A Bayesian interpolation method is further applied to the global compilation to provide interpolated fields as well as globally averaged B-Atm values. By performing a suite of experiments with a model of intermediate complexity, the authors further assess whether the reconstructed changes in deglacial B-Atm are due to ventilation changes or atmospheric radiocarbon dynamics.

The manuscript presents a unique and useful data compilation and tackles an important problem, i.e. deglacial changes in the carbon cycle. It is of high scientific quality and well written.

1) My main comment is related to the presentation of the modelling work and of the some results. I am puzzled by Figure 9. I find it very surprising to see such a different CO2 response across the different background conditions. For example, at BA assuming the symbol gets smaller as the forcing gets stronger, there is a linear decrease in CO2 as the SO winds decrease. However, no such relationship exists at 41ka and the LGM. Can the authors explain this? Maybe the colors have been mixed up? In addtion, please clearly state which symbol is which (size).

We have updated the figure caption to clarify that the smallest symbols refer to the control run, and that increasing sizes indicate increasing parameter reductions.

In Figure 9, the range of $CO_2$ sensitivities for the various processes used to drive changes in ventilation in the 'deglacial scenario', is broadly as illustrated in the original Figure 10 based on PI runs with 2,000 yrs of equilibration. As discussed in the manuscript, this broad range of sensitivity is encapsulated in the +/- 50% range that is applied to the theoretical 2-box model sensitivity (red lines in the original Figure 10).

However, as correctly noted by the Reviewer, divergent behaviour does emerge for Southern Ocean winds. Here. the final sensitivity depends primarily on the timescale of equilibration, rather than the background state. We alluded to this on line 541 of the original manuscript, but without delving more deeply into the issue. We have now remedied this by expanding this discussion (lines 548-556 in the revised text) and providing a supplementary figure that illustrates PI sensitivity tests for different equilibration times (new Figure S1).

The complexity of the response to Southern Ocean winds arises due to the intersection of impacts from: 1) wind-driven large-scale ocean overturning (and the relative dominance of northern- vs. southern deep-water masses); 2) shallow ocean mixing and residence times; and 3) gas-exchange/piston velocities. The first of these effects is 'slow', compared to the others. The key point, and the reason for the lack of a clear 'aging' with decreasing Southern Ocean wind strength in the 42ka and LGM snapshots of the original Figure 9, is

that an increased contribution of relatively young North Atlantic deep water to the deep Atlantic, with reduced preformed ages in the Southern Ocean, dominate the global average signal for longer equilibration times (>~2,000 years). The 42ka and LGM snapshots represent the effects of >10,000 years of equilibration with reduced Southern Ocean winds, where the Atlantic eventually gets 'younger' due to North Atlantic deep water (NDW) replacing Southern sourced deep water (SDW), and SDW leaving the surface ocean with reduced ages. The BA snapshot picks up a similar scaling to the 2000yr PI runs shown in original Figure 9, because for this snapshot the winds have just started to ramp up again to the PI parameterisation (leaving only ~2000yrs of equilibration), albeit with the opposite impact on 'ventilation' from winds than might have been expected.

I don't think it is necessary to display Table 2 in the main manuscript. There is a lot of information there, that should be simply conveyed in (an improved) Fig. 9. In addition, INT, FIX and CONST need to be defined in the caption of Table 2.

This is a helpful suggestion: we have moved Table 2 to the appendix (now Table S1), and defined the model runs in the revised caption.

Fig. 10a: I find it a bit surprising to only use the PI sensitivity experiments in that figure, when the previous figure apparently showed a different sensitivity of the DpCO2/D14C as a function of background state. I suppose using the BA would be more appropriate. I am also confused as to the necessity of the PI simulations.

As noted above, it is not so much the background climate state that matters in these idealised runs, but rather the timescale of equilibration. Furthermore, this only seems to apply to changes in Southern Ocean winds. Similar sensitivities are recovered for the other drivers regardless of equilibration time.

The PI sensitivity tests are 'tidier', in the sense of including fewer boundary condition changes. Indeed, using the 'snaphots' from the transient scenarios to illustrate the range of sensitivities, would require us to first remove the effects of changing 'climate' (temperature, salinity, etc.), as well as the 'attenuation biases' that arise due to implicit production rate changes (illustrated in the original Figure 9). This was the reason for performing the PI runs.

Therefore, we propose to retain the PI runs (with 2000 yrs equilibration time) in Figure 10 (Figure 11 in the revised manuscript). However, as noted above, we provide additional context for these, supported by additional PI runs with 5000yrs equilibration time included in the Appendix.

2) Conclusions

I find the conclusions a bit unclear. From the abstract L. 26-28, and L. 555-557, it is mentioned that "evolving ocean-atm exchange can account for 1/3 of the total GIG CO2 rise". Most studies suggest that the terrestrial carbon reservoir was smaller at the LGM than during the Holocene. This implies that the deglacial CO2 rise was due to a decrease in oceanic carbon, which in simple terms implies regional increase in CO2 outgassing and/or

reduced CO2 ocean uptake…. therefore ocean-atm exchange… what am I missing here? What do the authors mean?

Yes, this is a good point: we have clarified that we are referring to ocean 'ventilation' specifically (we were trying to avoid suggesting that 'ventilation' refers to transport changes alone, as is often assumed).  The relative contributions of export productivity changes (e.g. iron fertilisation), ventilation effects, and carbonate/alkalinity effects (or indeed volcanism etc.), remain to be quantified, after decades of research on this topic. In this context, we believe that our tentative quantification of the contribution from ventilation effects is quite significant, particularly in light of recent studies proposing that there was no change in ocean ventilation at all between the LGM and the PI.

It is further suggested that "half of the CO2 rise appears to have been associated with the BA". It is a confusing statement, particularly in the abstract as lacking context. Can the authors more precisely state what they mean?

We agree that this is confusing, in particular when dealing with averaged time-slices.  We have therefore changed this line to the hopefully clearer statement: "*[The ventilation] contribution to CO$_2$ rise appears to have continued through the Younger Dryas, though much of the impact was likely achieved by the end of the Bølling-Allerød, indicating a key role for marine carbon cycle adjustment early in the deglacial process.*"  We have tried to correct similar ambiguities throughout the revised text.

3) Minor edits:

L. 86-88: I do not understand the logic in that sentence.

We have tried to simplify the sentence.  The inferences of the Rafter et al. (2022) are perhaps a little confusing, but it is indeed the case that their study focussed on deglacial *transport* effects on B-Atm offsets. They inferred an increase in North Pacific ventilation at the LGM, despite also inferring decreased deep Pacific transport overall at this time.

L. 241: "and" did not consider?

Yes, corrected.

L. 253-256: This sentence is confusing.

We have tried to clarify.

L. 487: Figure 9 does not seem to suggest that the biases are attenuated for large changes in ventilation (i.e. no bias change as SO winds are decreased, even if colors/symbols have been mixed up).

The idealised simulations in Figure 9 cannot show the attenuation of such biases, as these simulations included prescribed atmospheric radiocarbon. We have added a note indicating this.

Fig. 8: Please provide the latitudinal boundaries used to define North Atlantic, North Pacific and Southern Ocean.

These have been added to the caption.

---

## Author Comment (AC2)

**Reviewer 2:**

In this manuscript, authors Skinner et al. perform several tasks that are of importance for the understanding of deglacial changes in atmosphere and ocean Δ14C: 1) By using an interpolation method, they produce gridded three dimensional fields of radiocarbon ages in the global ocean for different time slices associated with the last deglaciation. They then use models to attribute those changes and their relationship with air-sea CO2 transport to different processes in the ocean interior, and discuss the implications fro atmospheric Δ14C. The paper is clearly written and the figures are adequate. Follow some comments:

We are grateful to Juan Muglia for his detailed reading of the manuscript and for all the very helpful comments and corrections provided.

Major comment:

The only major comment I have is regarding the concept of transport rates governing ocean Δ14C. Throughout the manuscript, transport rate is discussed as a factor governing atmospheric-benthic Δ14C offsets. In the paper Muglia and Schmittner (2021), an analysis is performed with an ensemble of LGM model simulations. They find that mean ocean radiocarbon ages are much more closely related with deep ocean water mass structure than with overturning transport (please see Figs. 5 and 6 in that paper). I believe the authors should consider changing the attribution of Δ14C changes to deep water mass transport to deep water mass structure, and reflect that in the final version of their paper.

Yes, water mass 'geometry' is clearly a determinant of spatial B-Atm distributions, as it combines the influences of transport time and trajectory (i.e. transit time, gas-exchange and mixing history). We have added a note of this from line 452 in the revised text, where we also cite the study of Muglia & Schmittner (2021).

Please also note that while we do discuss transport rates as one factor influencing B-Atm offsets, we also emphasize that it is only one of several factors (e.g. from line 373 in the revised text). Indeed, our study is at pains to underline the dominant role of air-sea gas exchange in some aspects of deglacial marine radiocarbon (e.g. from line 421 in the revised text), while further noting and quantifying the additional influence of attenuation biases.

Minor comments:

Lines 60-70: Please include the values and uncertainties (if available) of ocean-atmosphere radiocarbon age offsets calculated by the cited literature.

These have been added to the revised text.

Line 320: "A few data points". Imprecise. Say the number of points.

This has been removed. We have updated the compilation to include a new study from the deep Indian Ocean that renders the 'Indian variant' exercise redundant. We have

therefore removed the 'Indian variant' and replaced it with a 'high sedimentation rate' data flag scenario, where only sites with sedimentation rates >10cm/kyr are retained.

Line 324: "This comparison highlights the Indian basin as an important target for future work". What type of future work? Please specify.

Added (we meant more reconstructions of past B-Atm offsets in the Indian basin).

Line 327: I don't understand the correlation coefficients expressed here. Are you calculating a correlation coefficient between data and a gridded interpolation calculated from the same data? If that is the case, what is the purpose of such calculation?

Yes, this is indeed what we have stated. The purpose of these correlation coefficients is to indicate how close the interpolation is able to get to the observations on average (if the correlation was poor, it would mean that the interpolation was only weakly guided by the data), bearing in mind that we use a Bayesian approach that strikes a balance between fitting each data point, and matching the volumetric representativity of all data locations in the modern ocean simultaneously.

Line 337: I can't use Figure 5 to compare with LGM with the modern state because the modern state is not plotted.

Yes, good point. We have decided to move the time-slice reconstructions for the HOL and EHOL from the Appendix to the main text, and therefore add a new figure that compares these with the BA and LGM, thus demonstrating the relative range of variability before and after the BA.

Figure 8: d14O? You probably mean d18O.

Yes, this has been corrected!

Line 1174: Typo "indicate are for constant"

Yes, corrected.

Figure 8: Please specify how the splines where calculated. What data did you use? Did you calculate them from time slices or using the x-axis of the age models?

This has been added to the caption; the splines use all available data, on their corrected age models, taking into account B-Atm uncertainties and the 'baseline' data flags, as described in the Methods section.

Figures 9 and 10: Please us the same color scheme for the experiments in these two figures.

This has been corrected.

Data availability comment:

The production of 3-dimensional past global fields of Δ14C based on data interpolation is very useful for the paleoceanography community. It will be good if the authors make those fields available on a repository.

These will be included in our data submission to PANGEA.